# DETECT EVERY THING WITH FEW EXAMPLES

## ABSTRACT

Few-shot object detection aims at detecting novel categories given a few example images. Recent methods focus on finetuning-based strategies to learn features representing novel classes, whose complicated procedures prohibit a wider application. In this paper, we introduce DE-ViT, a few-shot object detector without the need for finetuning. We transform the multi-class classification into multiple binary classifications, so a binary classifier can be trained and used for all classes without finetuning. We propose a novel propagation-based localization mechanism upon frozen DINOv2. We evaluate DE-ViT on few-shot, and one-shot object detection benchmarks with COCO and LVIS. For COCO, DE-ViT surpasses the few-shot SoTA by 15 mAP on 10-shot and 7.2 mAP on 30-shot and one-shot SoTA by 2.8 AP50. For LVIS, DE-ViT outperforms few-shot SoTA by 20 box APr. When compared to open-vocabulary detectors, DE-ViT outperforms the COCO SoTA by 6.9 AP50 and achieves 50 AP50 in novel classes, and surpasses LVIS SoTA by 1.5 mask APr and reaches 34.3 mask APr.

## 1   INTRODUCTION

Object recognition and localization are two of the core tasks in computer vision. *Few-shot object detection* provides a promising paradigm for generic object detector by representing novel categories with a set of support images (Antonelli et al., 2022). However, despite the promising practicality in principle, there still exist fundamental limitations in previous work on few-shot object detection. First, most recent few-shot methods rely on finetuning to adapt to novel classes (Köhler et al., 2023), a complicated and tedious procedure that restricts the practical use of these methods (Zhao et al., 2022). Second, the accuracy of existing few-shot methods does not keep up with other alternative solutions. Specifically, few-shot detectors fall behind open-vocabulary zero-shot detectors, especially in challenging datasets such as COCO and LVIS (Ma et al., 2023; Wu et al., 2023).

To alleviate the above limitations, we observe that most recent work represents novel categories with features produced from few-shot training (Köhler et al., 2023). However, without representation learning, few-shot training may not produce strong enough features. This could hinder few-shot performance. Motivated by this, we propose to build a few-shot detector on top of DINOv2 (Oquab et al., 2023), a strong pretrained vision model. To avoid finetuning over novel classes, we transform the multi-class classification into multiple binary classifications. Thus, a single binary classifier can be trained and used for all classes without any finetuning on novel classes. To accurately localize objects from frozen DINOv2 features, we design a novel propagation-based mechanism that localizes each object by propagating a region on the similarity map between DINOv2 features and class prototypes. Prototypes are class representatives built from support image features. With these proposed techniques, we introduce DE-ViT , a few-shot object detector that uses example images to detect novel objects without the need for finetuning. The overall architecture is illustrated in Fig. 2. A demonstration of detecting YCB objects is shown in Fig. 1.

We evaluate DE-ViT on few-shot, and one-shot object detection benchmarks with COCO (Lin et al., 2014) and LVIS (Gupta et al., 2019) datasets. Our method establishes new state-of-the-art (SoTA) results on all benchmarks. For COCO, DE-ViT surpasses the few-shot SoTA LVC (Kaul et al., 2022) by 15 mAP on 10-shot and 7.2 mAP on 30-shot and one-shot SoTA BHRL (Yang et al., 2022) by 2.8 AP50. For LVIS, which has been regarded as a highly challenging dataset for few-shot object detection (Wang et al., 2020), DE-ViT outperforms the SoTA DiGeo (Ma et al., 2023) by 20 box APr. When compared to open-vocabulary detectors, DE-ViT outperforms the COCO SoTA CORA⁺ (Wu et al., 2023) by 6.9 AP50 and reaches 50 AP50 and LVIS SoTA Ro-ViT (Kim et al.,

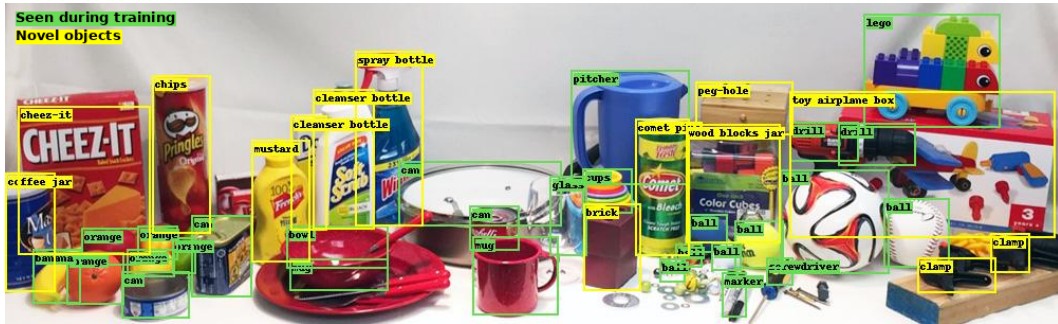

Figure 1: Demonstration of the proposed method on YCB objects (Calli et al., 2015). DE-ViT with ViT-L/14 is used for prediction. Note that our model is trained on only the base categories of LVIS. Example images of YCB objects are provided only during inference to represent novel categories.

2023) by 1.5 mask APr, reaching 34.3 mask AP in novel categories. Notably, our method only trains on the corresponding dataset, i.e., COCO or LVIS, without leveraging extra datasets for training, or distillation. Our contributions are summarized as follows: (1) To the best of our knowledge, we are the first to incorporate DINOv2 into solving the few-shot object detection problem. (2) Built upon the techniques motivated above, we introduce DE-ViT, which detects novel objects without any finetuning. (3) We demonstrate that DE-ViT establishes state-of-the-art performance over few-shot and one-shot benchmarks on COCO and LVIS, and outperforms open-vocabulary detectors as well.

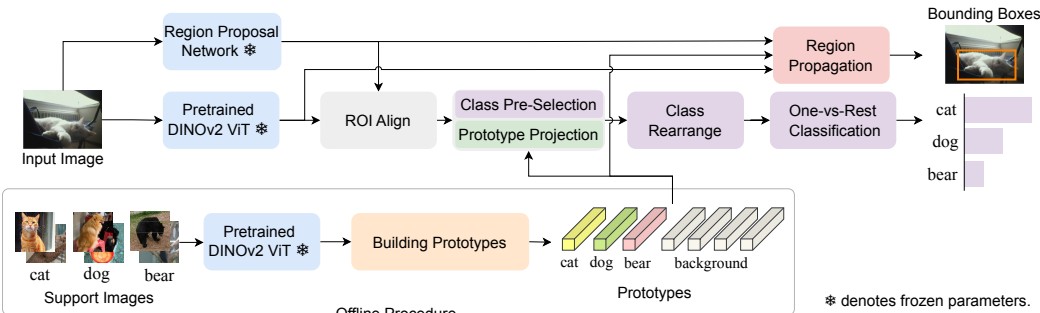

Figure 2: Overview of the proposed method. Our approach uses DINOv2 ViT to encode the image into a feature map, from which proposal features are extracted using ROIAlign. Proposals are generated via an off-the-shelf RPN. Prototype projection transforms proposal features into similarity maps based on prototypes derived from ViT features of support images. Multi-class classification of proposals is recast as a series of one-vs-rest binary classification tasks without the need for costly per-class inference. Refined localization is accomplished by our novel region propagation module. Both classification and refined localization rely exclusively on the computed similarity maps.

## 2 RELATED WORK

**Few-shot Object Detection (FSOD)** aims at detecting objects of novel classes by utilizing a few support images from novel classes as training samples (Köhler et al., 2023). Existing approaches can be broadly classified into finetuning-based (Wang et al., 2020; Fan et al., 2021; Sun et al., 2021; Xiao et al., 2022; Guirguis et al., 2023) and meta-learning-based strategies (Yan et al., 2019; Kang et al., 2019; Fan et al., 2020). Finetuning-based methods, despite their prevalence, suffer from a large accuracy gap between the base and novel classes, as well as practical limitations due to redundant multi-stage procedures (Zhao et al., 2022). Meta-learning methods avoid finetuning by online adaptation, but exhibit inferior accuracy (Köhler et al., 2023). One-shot Object Detection (OSOD) is an extreme case of FSOD with only one exemplar per novel class and simplifies the setting to single-class detection without finetuning (Yang et al., 2022). Prior approaches primarily focus on designing interaction mechanisms of dense spatial features between support and target

images (Antonelli et al., 2022). However, the OSOD formulation restricts the use of additional support and requires a separate inference per class. Compared with existing work, our method does not use finetuning or per-class inference and only utilizes class-level prototypes without dense feature interactions while outperforming SoTA methods in both few-shot and one-shot settings.

**Open-Vocabulary Object Detection (OVD)** aims at detecting objects of novel classes with only their category names and without support images (Zareian et al., 2021; Yu et al., 2023). However, despite this more challenging zero-shot setting, with the recent advances in vision-language models (Radford et al., 2021), open-vocabulary detectors outperform few-shot ones in COCO and LVIS (Ma et al., 2023; Wu et al., 2023). Our few-shot detector outperforms the SoTAs on both few-shot and open-vocabulary, which has not been achieved before by existing few-shot work.

## 3 METHOD

The major challenge in both FSOD is to generalize to classes that are unseen during training. However, despite numerous attempts to this issue, *e.g.*, margin-based regularization (Ma et al., 2023), there persists a considerable accuracy gap between base and novel classes. This disparity indicates that a network trained with base classes would inevitably fixate on patterns that are only present among a few base classes, which does not align with the objective of detecting arbitrary classes.

To prevent overfitting base class patterns, we propose to use the maps of similarities between the features and prototypes as the detector input. Thus, the network can only make decisions upon relevant information projected by the class representative features. Specifically, let $\mathbf{f} \in \mathbb{R}^{H \times W \times D}$ be the features of an image where $D$ represents the channel dimension and $(H, W)$ represent spatial dimensions, let $\mathbf{p} \in \mathbb{R}^{(C+B) \times D}$ be the prototypes where $C$ is the number of classes and $B$ is the number of class-agnostic background prototypes. Similarity map $\mathbf{s} \in \mathbb{R}^{H \times W \times (C+B)}$ is calculated using Eq. 1. This procedure is referred to in Fig. 3 as prototype projection.

$$\mathbf{s} = \mathbf{f} \cdot \mathbf{p}^\top \tag{1}$$

We adopt a standard two-stage object detection framework, *e.g.*, Mask R-CNN (He et al., 2017), which detects objects through RPN and RCNN stages. Existing literature has demonstrated that class-agnostic RPN proposals generalize well to novel classes (Gu et al., 2021). We use off-the-shelf RPNs to propose object regions and extract proposal features from DINOv2 ViT backbones. Similarity maps are computed between proposal features and prototypes, which are then fed to our architectures for classification and refined localization. Prototypes are constructed offline from support images with the procedures detailed in Sec. 3.3. The ViT backbones, prototypes, and resulting the similarity maps are kept frozen during detector training.

### 3.1 CLASSIFICATION WITH AN UNKNOWN NUMBER OF CLASSES

Unlike supervised learning, the number of classes in FSOD is indeterminate. The common strategy in FSOD is to extend the final linear layer with finetuning. In contrast with these existing approaches, we transform the multi-classification of $C$ classes into $C$ one-vs-rest binary classification tasks. In doing so, a single binary classifier could be trained and used for all classes without finetuning. However, applying a binary classifier for multi-class classification requires separate inference for each class (Zang et al., 2022). To avoid the costly per-class inference, we apply a class preselection to only predict the probabilities for a small selection of classes. Let $\bar{\mathbf{f}} = \frac{\sum_{i,j=1}^{H,W} \mathbf{f}_{ij}}{HW} \in \mathbb{R}^D$ be the average feature of a proposal. The class pre-selection procedure returns the top-$K$ mostly likely classes $\mathcal{C}_K = \text{top\_indices}_K(\mathbf{h})$, where $\mathbf{h} \in \mathbb{R}^C$ is the dot-product similarity between $\bar{\mathbf{f}}$ and class prototypes, defined as $\mathbf{h} = \bar{\mathbf{f}} \cdot \mathbf{p}^\top$. Our method only predicts the probabilities for $\mathcal{C}_K$ and sets the others to 0. As shown in Sec. 4.2, our method surpasses SoTA even when $K = 3$ on both COCO (80 classes) and LVIS (1203 classes), eliminating the need for costly per-class inference.

For each class $c_k$ in $\mathcal{C}_K$, the similarity map $\mathbf{s} \in \mathbb{R}^{H \times W \times (C+B)}$ (computed in Eq. 1) is rearranged into a class-specific map $\mathbf{s}_{c_k} \in \mathbb{R}^{H \times W \times (1+T+B)}$, as shown in Eq. 2, where $[C] \backslash c_k$ is defined as $\{1, .., c_k - 1, c_k + 1, ..., C\}$.

$$\mathbf{s}_{c_k} = \text{concat}(\mathbf{s}[:, :, c_k], F_{\text{rearrange}}(\mathbf{s}_{[C] \backslash c_k}), \mathbf{s}[:, :, C : C + B]) \tag{2}$$

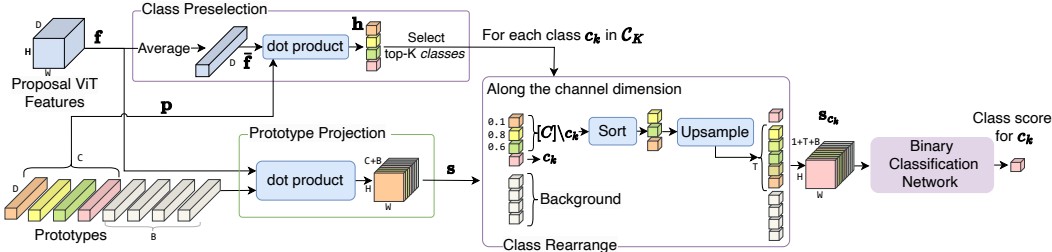

Figure 3: Overview of our classification architecture. Class pre-selection chooses the top-$K$ classes based on the dot product similarity between the average feature of each proposal and class-level prototypes. The probability of each selected class $c_k$ is predicted through a binary classification network, shared by all the classes, in a one-vs-rest manner. The input to this classification network is the similarity map that results from the prototype projection, after rearranging it for each class.

where $\mathbf{s}_{[C]\backslash c_k} = \text{concat}(\mathbf{s}[:,:,:c_k-1], \mathbf{s}[:,:,c_k+1:])$, and $F_{\text{rearrange}}(\mathbf{x})$ is defined in Eq. 3. $T$ is a constant hyper-parameter. In Eq. 2, $\mathbf{s}[:,:,c_k]$, $\mathbf{s}_{[C]\backslash c_k}$, and $\mathbf{s}[:,:,C:C+B]$ represent the similarity map for the current class $c_k$, other classes $[C]\backslash c_k$, and the background, correspondingly.

$$F_{\text{rearrange}}(\mathbf{x}) = \begin{cases} \text{upsample}(\text{sort}(\mathbf{x}), T) & \text{if } T \geq C-1 \\ \text{sort}(\mathbf{x})[:,:,:T] & \text{otherwise} \end{cases} \tag{3}$$

However, it is difficult to input $\mathbf{s}_{[C]\backslash c_k}$ directly to network due to the lack of inherent order in classes and the non-determinate nature of the number of classes in the few-shot setting. For the first issue, we decide to use magnitude order by sorting $\mathbf{s}_{[C]\backslash c_k}$ along the channel dimension, as in Eq. 3. For the second issue, we standardize the input size by either keeping the similarity of the top $T$ classes or linearly upsampling the similarity map of all $C-1$ classes to $T$. In doing so, $\mathbf{s}_{[C]\backslash c_k}$ is transformed from $H \times W \times (C-1)$ into a fixed size $H \times W \times T$. Note that the listed functions, *i.e.*, sort, concat and upsample, are applied along the channel dimension, and sort works in descending order. Finally, the class-specific similarity map $\mathbf{s}_{c_k}$ is given as input to a binary classification network that returns the probability for $c_k$. The overall classification architecture is illustrated in Fig. 3.

## 3.2 LOCALIZATION WITH REGION PROPAGATION.

Despite their rich semantics information, ViT features lack the coordinates information required for bounding box regression. As shown in Sec. 4.3, naively applying a conventional regression on ViT features yields poor localization results. A natural solution is to learn this localization capability by finetuning the ViT backbone during detector training. Similar strategies have demonstrated success in OVD for vision-language backbones, *e.g.*, CLIP (Zhong et al., 2022). However, we observed that finetuning results in an accuracy collapse on novel classes when text embeddings are replaced with prototypes, which indicates a loss of generalization power. While this phenomenon is intriguing, the practical question is how to produce accurate localization using frozen visual features.

Our intuition is that only proposals that overlap with objects are important because others would be rejected by the classification network. If we expand such proposals, they would possibly cover the entire region of underlying objects. Therefore, original proposals can be refined by predicting object regions within the expanded proposals. We model this procedure as propagating original proposals to object areas in the form of heatmaps. The heatmaps can be projected into box coordinates through integral over spatial dimensions. The overall refined localization architecture is illustrated in Fig. 4.

As shown in Fig. 4, the propagation procedure is implemented through binary segmentation. We convert groundtruth bounding boxes to heatmaps that we use to train this segmentation network. Inspired by unsupervised keypoint estimation, particularly the works of IMM (Jakab et al., 2018) and Transporter (Kulkarni et al., 2019), we devise a spatial integral layer to project the propagated heatmap to box. The idea is to learn a transformation that translates the heatmap to a box $(c_w^{\text{rel}}, c_h^{\text{rel}}, w^{\text{rel}}, h^{\text{rel}}) \in [0,1]^4$ in coordinates that are relative to the expanded proposal. Relative coordinates can be simply mapped back to absolute coordinates. Let $\mathbf{g} \in \mathbb{R}^{H \times W}$ denote the logits of the propagated heatmap, the relative box $(c_w^{\text{rel}}, c_h^{\text{rel}}, w^{\text{rel}}, h^{\text{rel}})$ can be estimated by our spatial integral

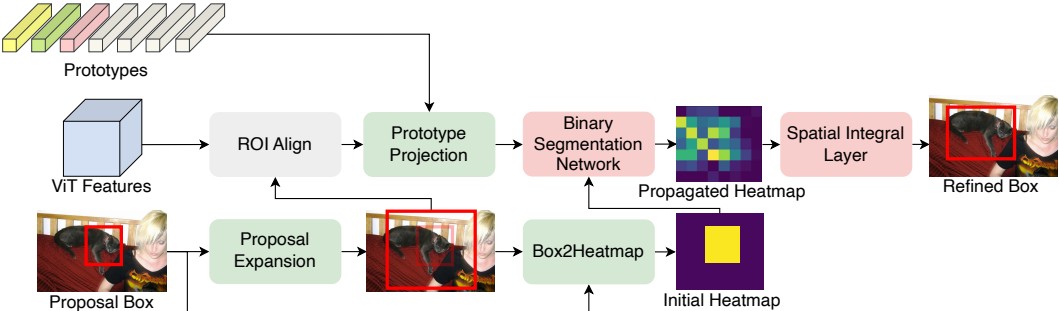

Figure 4: Overview of our refined localization architecture. Proposal expansion enlarges each proposal by a fixed ratio to cover more object area. The spatial relationship between the original and expanded proposal is described via a heatmap. The segmentation network navigates the initial heatmap toward accurate object regions. The propagated heatmap is converted into bounding box coordinates through our spatial integral layer.

layer as explained in Eq. 4 and 5. An illustrative example of our spatial integral layer can be found in Figure 5.

$$(c_w^{\text{rel}}, c_h^{\text{rel}}) = \sum_{i,j=1}^{H,W} \left( \frac{i}{H}, \frac{j}{W} \right) * \text{softmax}(\mathbf{g})_{ij} \tag{4}$$

$$(w^{\text{rel}}, h^{\text{rel}}) = \left( \sum_{i=1}^{H} \sum_{j=1}^{W} \frac{\sigma(\mathbf{g})_{(i)j}}{W} \theta^{\mathbf{w}}{}_i, \sum_{j=1}^{W} \sum_{i=1}^{H} \frac{\sigma(\mathbf{g})_{i(j)}}{H} \theta^{\mathbf{h}}{}_j \right) \tag{5}$$

To motivate Eq. 4 and 5, consider the toy example of converting a binary mask to a bounding box in Fig. 5. A reasonable approach is to compute the mask center as the bounding box center and pick the maximum row and column sum as width and height. Following the same spirit, we compute the expected position under the spatial distribution $\text{softmax}(\mathbf{g})$ as the bounding box center. We compute the row and column sums of sigmoid activation as $\sum_{j=1}^{W} \sigma(\mathbf{g})_{ij}$ and $\sum_{i=1}^{H} \sigma(\mathbf{g})_{ij}$. Instead of picking the maximum, we aggregate all row or column sums in terms of magnitude. The rationale of this aggregation is that a larger estimation is more likely to over-cover the entire object and a small estimation may produce a box too tight. The aggregation is done by sorting the estimation and then weighted averaging. This explains the use of

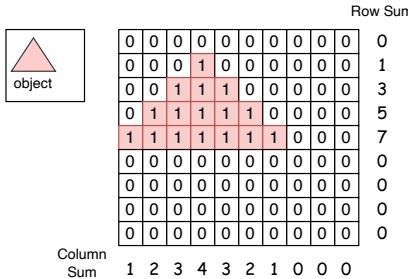

Figure 5: To compute the bounding box of the triangle object from the binary mask, the maximum row and column sum can be used as the box width and height.

order statistics notation $(i)$ and $(j)$, and $\theta^{\mathbf{h}} \in \mathbb{R}^W, \theta^{\mathbf{w}} \in \mathbb{R}^H$ are learnable aggregation weights. Finally, the relative coordinates are mapped to absolute ones by Eq. 6, where $(c_w^{\text{exp}}, c_h^{\text{exp}}, w^{\text{exp}}, h^{\text{exp}})$ is the expanded proposal.

$$\begin{aligned} (w^{\text{out}}, h^{\text{out}}) &= (w^{\text{exp}} \, w^{\text{rel}}, h^{\text{exp}} \, h^{\text{rel}}) \\ (c_w^{\text{out}}, c_h^{\text{out}}) &= (c_w^{\text{exp}} - 0.5 w^{\text{exp}}, c_h^{\text{exp}} - 0.5 h^{\text{exp}}) + (c_w^{\text{rel}} \, w^{\text{exp}}, c_h^{\text{rel}} \, h^{\text{exp}}) \end{aligned} \tag{6}$$

During training, we use groundtruth bounding boxes as regression targets for the output of spatial integral layer. Similar to our classification pipeline, the localization procedure is applied for each class in $\mathcal{C}_K$ to produce class-specific boxes. For each class $c_k$, only the similarity map of $c_k$ and the background will be selected as input. We omit this detail in Fig. 4 for visual clarity.

### 3.3 BUILDING PROTOTYPES

Similar to the pioneering FSOD work Meta R-CNN (Yan et al., 2019), our method represents classes with prototype vectors constructed from visual features of given support images. The main differ-

Table 1: Results on COCO 2014 few-shot benchmark. Our method outperforms existing work in detecting novel classes by a significant margin and does not require finetuning on novel classes.

| Method | Finetune on Novel | 10-shot | | | | 30-shot | | | |
|---|---|---|---|---|---|---|---|---|---|
| | | bAP | nAP | nAP50 | nAP75 | bAP | nAP | nAP50 | nAP75 |
| FSRW (Kang et al., 2019) | ✗ | - | 5.6 | 12.3 | 4.6 | - | 9.1 | 19 | 7.6 |
| Meta R-CNN (Yan et al., 2019) | ✗ | 5.2 | 6.1 | 19.1 | 6.6 | 7.1 | 9.9 | 25.3 | 10.8 |
| TFA (Wang et al., 2020) | ✓ | 33.9 | 10 | 19.2 | 9.2 | 34.5 | 13.5 | 24.9 | 13.2 |
| Multi-Relation Det (Fan et al., 2020) | ✗ | - | 16.6 | 31.3 | 16.1 | - | - | - | - |
| FSCE (Sun et al., 2021) | ✓ | - | 11.9 | - | 10.5 | - | 16.4 | - | 16.2 |
| Retentive RCNN (Fan et al., 2021) | ✓ | 39.2 | 10.5 | 19.5 | 9.3 | 39.3 | 13.8 | 22.9 | 13.8 |
| HeteroGraph (Han et al., 2021) | ✓ | - | 11.6 | 23.9 | 9.8 | - | 16.5 | 31.9 | 15.5 |
| FsDetView (Xiao et al., 2022) | ✓ | 6.4 | 7.6 | - | - | 9.3 | 12 | - | - |
| Meta Faster RCNN (Han et al., 2022a) | ✓ | - | 12.7 | 25.7 | 10.8 | - | 16.6 | 31.8 | 15.8 |
| LVC (Kaul et al., 2022) | ✓ | 28.7 | 19 | 34.1 | 19 | 34.8 | 26.8 | 45.8 | 27.5 |
| CrossTransformer (Han et al., 2022b) | ✓ | - | 17.1 | 30.2 | 17 | - | 21.4 | 35.5 | 22.1 |
| NIFF (Guirguis et al., 2023) | ✓ | 39 | 18.8 | - | - | 39 | 20.9 | - | - |
| DiGeo (Ma et al., 2023) | ✓ | 39.2 | 10.3 | 18.7 | 9.9 | 39.4 | 14.2 | 26.2 | 14.8 |
| DE-ViT (Ours)  ViT-S/14 | ✗ | 24 | **27.1** | **43.1** | **28.5** | 24.2 | **26.9** | **43.1** | **28.4** |
| ViT-B/14 | ✗ | 28.3 | **33.2** | **51.4** | **35.5** | 28.5 | **33.4** | **51.4** | **35.7** |
| ViT-L/14 | ✗ | 29.4 | **34.0** | **53.0** | **37.0** | 29.5 | **34.0** | **52.9** | **37.2** |

ence in building prototypes is that we use the features from pretrained DINOv2 while Meta R-CNN uses region features from its detection network. Fig. 6 shows the process of building instance-level prototypes. For each object instance, its prototype is computed as the mean ViT feature from corresponding regions defined by either a segmentation mask or bounding box. Next, class-representing prototypes are obtained by averaging the cluster centroids of instance-level prototypes for each class. We use the online clustering method proposed by SwAV (Caron et al., 2020), though we find that simply averaging all instance prototypes for each class achieves similar results. Prototypes of base classes are built from the entire training set instead of support images, which are used only for novel classes.

To build background prototypes, we start from the obervation that backgrounds typically share similar visual attributes, such as uniform motion, smooth texture, static color tone, *etc*. Moreover, the ability to capture and thereby separate background from foreground is crucial. Because of the lack of visual diversity yet the importance of background semantics, we

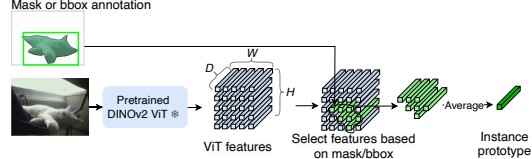

Figure 6: Overview of building instance prototypes.

use masks for a fixed list of background classes, *e.g.*, sky, wall, road, and apply a similar prototype-building procedure. The idea of using background information for few-shot performance is similarly explored in IPRNet (Okazawa, 2022). In contrast with class-level prototypes that change upon class configurations, background prototypes are always fixed as a part of the network's parameters. All the prototypes are built offline and frozen during training and inference.

## 4 EXPERIMENTS

We comprehensively evaluate our method on few-shot, one-shot benchmarks, and compare it to open-vocabulary detectors. Furthermore, we compare the efficiency of our method against SoTA solutions, study few-shot performance on different numbers of shots, and provide qualitative results. We conduct ablations to show that a naive combination of DINOv2 and Meta R-CNN leads to unsatisfying results, and every proposed component is important to the performance of our method.

**Evaluation Metrics and Datasets.** Few-shot, one-shot, and open-vocabulary evaluations split classes into base and novel classes. Base classes are seen during training and novel classes are unseen. The performance on novel classes is more important. For COCO, nAP, nAP50, and nAP75 represent mAP, AP50, and AP75 in novel classes. bAP and bAP50 represent mAP and AP50 in base classes. One-shot evaluation conventionally divides 80 classes of COCO into four even partitions, and alternatively takes three as base classes and one partition as novel classes (Michaelis et al., 2018). There are 4 base/novel splits in total, named Split-1/2/3/4. For LVIS, APr, APc, and APf represent AP on rare, common, and frequent categories. Rare categories are used as novel classes. Metrics on LVIS are computed separated on bounding boxes, *e.g.*, box APr, or instance segmenta-

Table 2: Results on COCO 2017 one-shot benchmark. DE-ViT outperforms existing work and is not limited to single class detection and single support image as other one-shot methods.

| | bAP50 | | | | | nAP50 | | | | |
|---|---|---|---|---|---|---|---|---|---|---|
| | Split-1 | Split-2 | Split-3 | Split-4 | Avg | Split-1 | Split-2 | Split-3 | Split-4 | Avg |
| SiamMask (Michaelis et al., 2018) | 38.9 | 37.1 | 37.8 | 36.6 | 37.6 | 15.3 | 17.6 | 17.4 | 17 | 16.8 |
| CoAE (Hsieh et al., 2019) | 42.2 | 40.2 | 39.9 | 41.3 | 40.9 | 23.4 | 23.6 | 20.5 | 20.4 | 22 |
| AIT (Chen et al., 2021) | 50.1 | 47.2 | 45.8 | 46.9 | 47.5 | 26 | 26.4 | 22.3 | 22.6 | 24.3 |
| SaFT (Zhao et al., 2022) | 49.2 | 47.2 | 47.9 | 49 | 48.3 | 27.8 | 27.6 | 21 | 23 | 24.9 |
| BHRL (Yang et al., 2022) | 56 | 52.1 | 52.6 | 53.4 | 53.6 | 26.1 | 29 | 22.7 | 24.5 | 25.6 |
| DE-ViT (Ours, ViT-L/14) | **59.4** | **57.0** | **61.3** | **60.7** | **59.6** | **27.4** | **33.2** | **27.1** | **26.1** | **28.4** |

tion masks, *e.g.*, mask APr. We evaluate our method on COCO 2014, COCO 2017 (Lin et al., 2014), and LVIS-v1 (Gupta et al., 2019). We follow the conventional base/novel classes split with existing work (Wang et al., 2020; Yang et al., 2022; Zhong et al., 2022). Note that the few-shot benchmark uses COCO 2014, while the one-shot and open-vocabulary benchmarks use COCO 2017.

**Model Specifications.** We use DINOv2 (Oquab et al., 2023) ViT as the feature extractor, and report results in ViT-S/B/L (small, base, large) model sizes. We train a ResNet50 RPN separately for each dataset using only base classes and use 1000 proposals in all settings. We detail the design of binary classification and segmentation networks in Sec. A.2. Class prototypes are built upon instance masks in support images unless specified. We use the same support images as sampled by previous work (Wang et al., 2020; Yang et al., 2022) for few-shot and one-shot settings. When comparing with open-vocabulary detectors, we sample 30 instances per class (30-shot) using the protocol of Wang et al. (2020) for the support set. Background prototypes are extracted from background classes, *e.g.*, sky, road, by the semantic masks in COCOStuff (Caesar et al., 2018).

## 4.1 MAIN RESULTS

Tab. 1 shows our results on few-shot COCO benchmark. DE-ViT outperforms the previous SoTA LVC by a significant margin (+15 nAP on 10-shot, +7.2 nAP on 30-shot). It is worth noting that LVC requires over ten stages for self-training and pseudo-labeling procedures (Kaul et al., 2022). A pretrained model for LVC has never been released. Other recent few-shot works also include multiple pretraining and finetuning stages (Han et al., 2022a;b). Our proposed method DE-ViT can be trained in a single stage and used on novel objects directly without any fine-tuning. The difference between 30-shot (nAP50=52.9) and 10-shot (nAP50=53.0) is smaller than 0.1% and could be interpreted as statistically insignificant.

LVIS has been regarded as a highly challenging dataset in FSOD (Wang et al., 2020) and only Di-Geo (Ma et al., 2023) reports few-shot results on LVIS v1. Tab. 3 shows that our method outperforms DiGeo in all metrics and a significant boost in the accuracy of detecting novel objects (+20 box APr).

Tab. 2 shows our results on one-shot COCO. DE-ViT outperforms the previous SoTA BHRL by 6 bAP50 and 2.8 nAP50. One-shot methods follow a single-class detection setting. We adapt DE-ViT by detecting each class separately during evaluation. It is worth noting that our proposed DE-ViT can perform both one-shot and few-shot detection using the same model. OWL-ViT (Minderer et al., 2022) also reports one-shot results on COCO. However, OWL-ViT's results are obtained with an ensemble of open-vocabulary and one-shot pipelines without providing implementation or isolated measurements. Therefore, we only compare against OWL-ViT with our few-shot setting in Tab. 4.

Table 3: Performance comparison with existing few-shot object methods on LVIS dataset. We report box AP as evaluation metrics.

| Method | | APr | APc | APf | AP |
|---|---|---|---|---|---|
| DiGeo (Ma et al., 2023) | | 16.6 | 22.8 | 28 | 24.4 |
| | ViT-S/14 | **23.4** | 22.8 | 22.5 | 22.8 |
| DE-ViT (Ours) | ViT-B/14 | **26.8** | **26.5** | 25.3 | **26** |
| | ViT-L/14 | **33.6** | **30.1** | **30.7** | **30.9** |

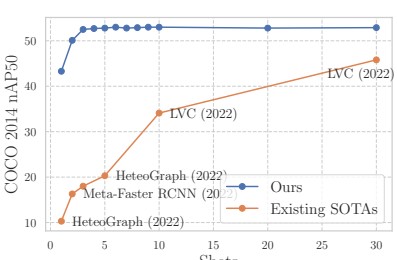

Figure 7: Performance of our method and few-shot SoTA at different shots.

Table 4: Comparison to open-vocabulary detectors on LVIS and COCO 2017. †: Customized pre-trained model is used instead of public models. "-" denotes result that is not reported.

| Method | Backbone | Use Extra Training Set | LVIS mask APr | LVIS box APr | COCO nAP50 |
|---|---|---|---|---|---|
| ViLD (Gu et al., 2021) | EffNet-B7 | ✗ | 26.3 | 27 | 27.6 |
| RegionCLIP (Zhong et al., 2022) | RN50x4 | ✓ | - | 22 | 39.3 |
| OV-DETR (Zang et al., 2022) | ViT-B/32 | ✗ | - | 17.4 | 29.4 |
| Detic (Zhou et al., 2022b) | RN50 | ✓ | 17.8 | - | 27.8 |
| OWL-ViT (Minderer et al., 2022) | ViT-L/14† | ✗ | - | 25.6 | - |
| OWL-ViT (Minderer et al., 2022) | ViT-L/14† | ✓ | - | 31.2 | - |
| CORA+ (Wu et al., 2023) | RN50x4 | ✓ | - | 28.1 | 43.1 |
| Ro-ViT (Kim et al., 2023) | ViT-L/14† | ✗ | 31.4 | - | 33 |
| F-VLM (Kim et al., 2023) | RN50x64 | ✗ | 32.8 | - | 28.0 |
| DE-ViT (Ours) | ViT-S/14 | ✗ | 24.2 | 23.4 | 39.5 |
| DE-ViT (Ours) | ViT-B/14 | ✗ | 28.5 | 26.8 | **45.4** |
| DE-ViT (Ours) | ViT-L/14 | ✗ | **34.3** | **33.6** | **50** |

We compare DE-ViT against open-vocabulary detectors in Tab. 4. DE-ViT outperforms the previous SoTA CORA+ by 6.9 AP50. Our method only trains on COCO while CORA+ uses ImageNet-21K (Krizhevsky et al., 2012) and COCO Captions (Chen et al., 2015) as additional training data. When only using COCO, DE-ViT outperforms CORA by 8.3 AP50. In LVIS, DE-ViT outperforms the previous SoTA on mask APr (+1.5 over F-VLM) and box APr (+2.4 over OWL-ViT). We follow a multi-scale instance segmentation head design, as detailed in Sec. A.2.

We observe a high variance in the performance of OVD detectors in Tab. 4. F-VLM achieves a mask APr of 32.8 on LVIS but only has 28 nAP50 on COCO. While CORA+ has 43.1 nAP50 on COCO but only a box APr of 28.1 on LVIS. On the contrary, our DE-ViT outperforms existing solutions on both LVIS (34.3 mask APr) and COCO (50 nAP50). We acknowledge the task setting difference of this comparison. Our few-shot method DE-ViT has access to support images that language-based detectors do not. However, we emphasize that DE-ViT outperforms the SoTAs on both few-shot and open-vocabulary, which has not been achieved before by existing few-shot work.

## 4.2 ANALYSIS

Table 5: Inference time comparison on COCO with existing few-shot methods.

| | Backbone | Secs/Img | nAP50 |
|---|---|---|---|
| Meta FasterRCNN (Han et al., 2022a) | RN101 | 0.61 | 31.8 |
| CrossTransformer (Han et al., 2022b) | Custom | 3 | 35.8 |
| DE-ViT (Ours) | ViT-S | **0.25** | **42.7** |

Table 6: Comparison of training epoch and parameter size to other detectors trained on LVIS.

| | Total Params | Trained Params | Epochs | APr |
|---|---|---|---|---|
| OWL-ViT (Minderer et al., 2022) | 433M | 433M | 1800 | 31.2 |
| F-VLM (Kuo et al., 2023) | 445M | 25M | 118 | 32.8 |
| DE-ViT (Ours) | 350M | **23M** | **14.4** | **34.3** |

**Efficiency.** We compare the inference time of DE-ViT against recent few-shot works in Tab. 5. Tab. 5 shows that DE-ViT has the smallest inference time, while having better accuracy. We compare the parameter sizes and training epochs of detectors trained on the large-scale dataset LVIS in Tab. 6. Tab. 6 shows that DE-ViT only has 23M trainable parameters, and is trained orders of magnitude faster than F-VLM (Kuo et al., 2023) and OWL-ViT (Minderer et al., 2022). The inference time of all methods is measured under the same machine. We conduct inference time comparisons under different values of $K$ and detail the efficiency discussion in Sec. A.1.

**More shots.** To study the model performance with different numbers of shots, we plot the nAP50 with different shots in Fig. 7 and Fig. A2 for COCO 2014 and COCO 2017, correspondingly. It can be seen that performance generally increases with the number of shots, and there exists an inflection point after which more samples do not help. The inflection point is located around 50 to 75 shots for COCO 2017 and 6 shots for COCO 2014. We detail the shot sampling setup in Sec. A.1.2.

**Qualitative Results.** We provide qualitative comparisons of our proposed DE-ViT and Meta Faster RCNN in Fig. 8 and Fig. A10. DE-ViT detects more novel objects while having fewer false positives. Note that Meta Faster RCNN can only detect novel objects after finetuning, while DE-ViT can detect both base and novel classes without finetuning. We provide more visualizations in Fig. A9 and A11.

Table 7: Ablation studies on the classification architecture.

| Conventional Prototype Head | Prototype Projection | Background Tokens | Class Rearrange | Novel | | Base | |
|---|---|---|---|---|---|---|---|
| | | | | nAP50 | nAP75 | bAP50 | bAP75 |
| ✓ | | | | 4.5 | 2.2 | 48.9 | 22.5 |
| | ✓ | | | 26.2 | 9.7 | 29.3 | 12 |
| | ✓ | ✓ | | 38.4 | 23 | 43.4 | 26.8 |
| | ✓ | ✓ | ✓ | 39.5 | 24.1 | 42.3 | 25.9 |

Table 8: Ablation studies on the propagation localization.

| Conventional REG Head | Expanded Proposal | Spatial Integral over Heatmap | Novel | | Base | |
|---|---|---|---|---|---|---|
| | | | nAP50 | nAP75 | bAP50 | bAP75 |
| ✓ | | | 37.7 | 14.6 | 46.5 | 23.9 |
| ✓ | ✓ | | 35.6 | 12.5 | 41.3 | 19.8 |
| | ✓ | ✓ | 39.5 | 24.1 | 42.3 | 25.9 |

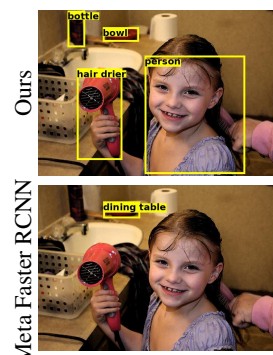

Figure 8: Qualitative comparison against Meta Faster RCNN. More visualizations can be found in Fig. A4, A9, A10, and A11.

### 4.3 ABLATION STUDY

**Naive Combination of DINOv2 and Meta RCNN.** Our initial attempt is to integrate DINOv2 into the framework of Meta RCNN (Yan et al., 2019), a classic prototype-based few-shot detector. Specifically, we build prototypes from DINOv2 features and use the prototype features as the weights for the last linear layer of the classification network. The only difference between the above procedure and Meta RCNN is how to build prototypes. We use DINOv2 features and Meta RCNN uses average features of the output from ROIAlign in a detection network. It is worth noting that this procedure is also similar to RegionCLIP which uses text embedding features to compose the weights of the last classification linear layer. As shown in Tab. 7, this conventional prototype head design produces almost no few-shot ability (4.5 nAP50). This suggests that combining DINOv2 and prototype-based few-shot detectors is not trivial.

**Classification.** We examine the component effects for classification in Tab. 7. Tab. 7 shows that the model completely overfits base classes without our proposed components. By learning from similarity maps from prototype projection, general detection ability emerges, and then improves after adding the background prototypes and the class rearrange module. Results in Tab. 7, 8 and A1 are obtained with ViT-S/14.

**Refined Localization.** We study the impacts of propagation-based localization in Tab. 8. For the conventional REG head, we use a CNN with both features and similarity maps as input. Novel classes have poor localization (14.6 AP75) without our proposed architecture, and simply expanding proposals even lowers the performance. The localization accuracy jumps to 24.1 AP75 when our proposed spatial integral layer is used to produce bounding boxes from the propagated heatmaps.

**Using Boxes or Masks to Build Prototypes.** In Tab. A1, we study the effects of annotation types, *i.e.*, bounding boxes or masks, in constructing prototypes. We observe that using bounding boxes to build prototypes yields almost indistinguishable performance compared to using instance masks at even 5-shot in COCO 2014. Our main results use prototypes built from instance masks.

## 5 CONCLUSION

In this work, we propose DE-ViT, a few-shot detector that uses example images to detect novel classes without any finetuning. We demonstrate that DE-ViT establishes new state-of-the-art in few-shot and one-shot benchmarks. DE-ViT also outperforms open-vocabulary detectors, which has not been achieved before by existing few-shot work. DE-ViT detects objects using only frozen DINOv2 features. Therefore, it is straightforward to integrate DE-ViT with backbones other than DINOv2. One of the limitations is that our current architecture is a mix of ViT and RCNN, while a full transformer network can clearly be more scalable and unlock more possible abilities and integrations to other modalities. Another limitation is that our current model relies on an external region proposal network (RPN). It is possible to train an RPN on top of frozen DINOv2 with some engineering efforts. We hope that our work will be useful in downstream tasks such as robotic manipulation, and help other researchers develop better methods for few-shot object detection.

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

# A APPENDIX

## A.1 ADDITIONAL EXPERIMENTS

### A.1.1 OVER-EXPANDED PROPOSAL ANALYSIS

Our propagation mechanism localizes objects by first expanding the proposals and then refining the initial bounding boxes within the expanded proposals. Here we study the localization behavior of the scenario where expanded proposals cover multiple same-class objects. We will use the term *over-expanded proposals* to denote this scenario.

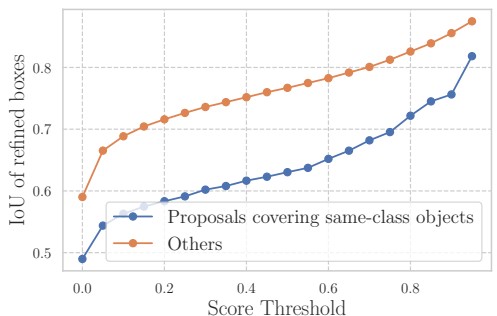

In Fig. A1, we plot the average IoU between refined boxes and groundtruth objects under different score thresholds. A higher IoU indicates more accurate localization. It can be seen that over-expanded proposals generally degrade localization accuracy, as their final predicted boxes have smaller IoU towards the ground truth. But the degradation is far from total, e.g., they still

Figure A1: IoU of refined bounding boxes with groundtruth objects under different score thresholds. The blue curve denotes refined boxes from over-expanded proposals.

produce boxes whose IoU > 0.7 on average, under the score threshold of 0.85, which we use to generate qualitative visualizations. We also observe that the over-expanded proposals occupy around 7% of all proposals in our model for COCO, and only half of them appear in the final prediction (after NMS and score filtering). This means even if an over-expanded proposal predicts an inaccurate box, its impact is softened by filtering of NMS and score thresholding.

In terms of actual behavior, we list success and failure cases of over-expanded proposals in Fig. A7 and A8. In failure cases, propagation either encompasses both objects or produces erratic boxes. But all failure cases happen under inferior proposals, where the initial proposals are already poorly located and do not cover any object, or cover multiple objects before expansion. In successful cases, propagation generally prefers central objects but can locate an object accurately regardless of the proposal quality. This means our propagation localization does not fully rely on the proposal quality.

### A.1.2 DETECTION ACCURACY AT MORE SHOTS

We study the model performance with different numbers of shots in Fig. 7 and A2, on COCO 2014 and COCO 2017, correspondingly. For COCO 2014, the numbers of shots are set from 1 to 10, 15, 20, and 30. To align with existing work, we use the same support images by previous work (Wang et al., 2020) for shots 2,3,5,10,30. For other shots, we sample within the support images mentioned above. For COCO 2017, we follow the convention of the one-shot benchmark, since there does not exist a few-shot standard split of base/novel classes and preselected support images. Specifically, we compute nAP50 for all four splits and randomly select support images within the validation set of COCO 2017 for each target image. The reported nAP50 is the average among all splits and choices of support images. The numbers of shots are set from 1 to 10, 15, 20, 30, 40, 50, 75, 100.

Table A1: Ablation studies on annotation types used to build prototypes.

| Support Images | nAP50 | | |
| Annotation | 5-shot | 10-shot | 30-shot |
| --- | --- | --- | --- |
| mask | 43.1 | 43.1 | 43.1 |
| bbox | 43 | 42.6 | 43.1 |

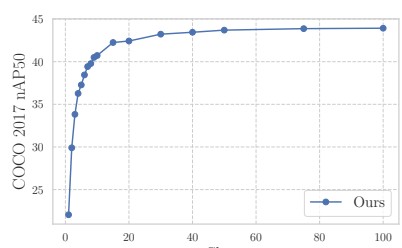

Figure A2: Few-shot detection accuracy under different numbers of shots in COCO 2017.

Table A2: Efficiency comparison against existing state-of-the-art on COCO OVD benchmark.

| Method | Backbone | top $K$ | Novel AP50 | Secs/Img |
|---|---|---|---|---|
| DE-ViT (Ours) | ViT-L/14 | 1 | **47.7** | **0.22** |
| | | 3 | **50.0** | **0.33** |
| | | 10 | **50.0** | 0.83 |
| CORA | RN50x4 | - | 41.7 | 0.5 |

Table A3: Efficiency comparison against existing state-of-the-art on LVIS OVD benchmark.

| Method | Backbone | top $K$ | box APr | Secs/Img |
|---|---|---|---|---|
| DE-ViT (Ours) | ViT-L/14 (pytorch) | 1 | 25.4 | 0.35 |
| | | 3 | **32.6** | 0.5 |
| | | 10 | **33.6** | 1.58 |
| OWL-ViT | ViT-L/14 (jax) | - | 31.2 | **0.42** |

### A.1.3 EFFICIENCY ANALYSIS

In Tab. A2 and A3, we compare the efficiency of DE-ViT against previous SoTA open-vocabulary methods with different values of $K$. $K$ is the number of classes ($K$) chosen to predict probability during class preselection. Our main results are obtained using $K = 10$ and the result in Tab. A2 is obtained using $K = 3$. The inference time of all methods is measured under the same machine with PyTorch 1.13 and a single A100 GPU. Similar to RegionCLIP (Zhong et al., 2022), DE-ViT requires a pretrained and frozen region proposal network. Recent works like F-VLM (Kuo et al., 2023) and FC-CLIP (Yu et al., 2023) propose to train detectors directly on top of a frozen backbone without an additional RPN. However, our region proposal network involves a small computation cost. In our model for LVIS, RPN has only 27M parameters, and the number of parameters for the ViT-L backbone is 320M. The F-VLM backbone (RN50x64) has 420M parameters. In our model for COCO, RPN has only 8M parameters.

As shown in Tab. A2 and A3, DE-ViT outperforms SoTA while being much faster at COCO and slightly slower at LVIS when $K = 3$. Note that OWL-ViT is implemented in JAX framework (Frostig et al., 2018). JAX is commonly known as faster than pytorch (Phan et al., 2019). We select CORA instead of CORA⁺ and OWL-ViT rather than F-VLM for comparison because CORA⁺ and the RN50x64 version of F-VLM have not been released at the moment this paper is written. Note that we use half-precision for ViT inference as the standard practice for vision transformers and full precision for other layers. We conduct inference time comparisons against open-vocabulary methods in addition to few-shot methods because of their superior accuracy.

### A.1.4 FEATURE VISUALIZATION OF DINOV2 AND CLIP

The critical distinction between DE-ViT and OVD methods is that DE-ViT uses visual features and represents each class by the center of visual features, while OVD methods represent each class with text features, mostly from CLIP. Therefore, we sample and visualize DINOv2 and CLIP features in Fig. A3 using UMap (McInnes et al., 2018). CLIP visual features also show excellent intra-class compactness. However, there is a huge vacuum between CLIP text and visual features, and text features are almost overlapped with each other. This indicates that the distances between text and visual features may be more susceptible to noises. This further suggests that using images to represent classes could be more promising than only using texts.

Visual features of DINOv2 are extracted using groundtruth bounding boxes with the procedure shown in Fig. 6. For CLIP, we follow the standard text and region visual features extraction procedure (Zhong et al., 2022; Wu et al., 2023). Specifically, visual features are obtained by applying ROIAlign on the output of `res4` block. The ROIAlign outputs are fed into `res5` and attention pooling blocks to produce the final feature vector. Text features are extracted with CLIP text encoders with category names. We randomly sample 30 instances per class for visualization, *i.e.*, each node (✖) represents an instance of the corresponding class. The UMap dimension reduction transformation is learned with a larger set of instances with both visual and text features.

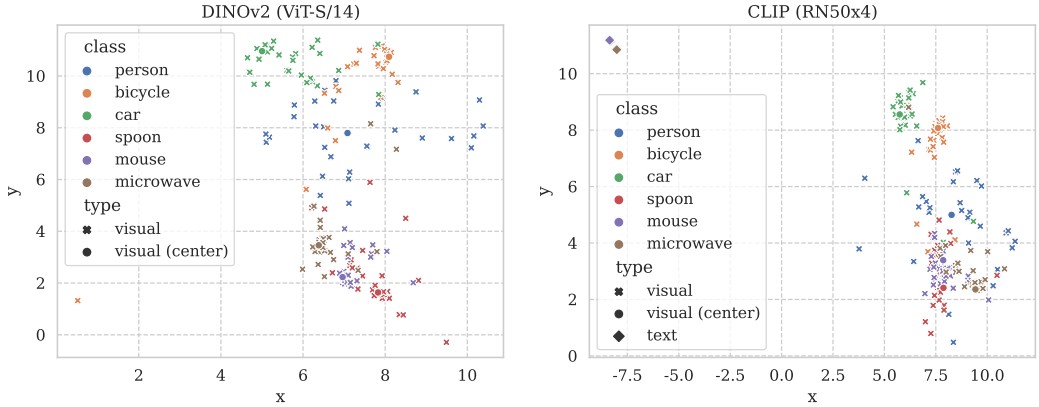

Figure A3: Visualization of DINOv2 and CLIP (RN50x4) features in 2d plane. From visual features (✖), both DINOv2 and CLIP show good concentration within each class. However, CLIP text features (◆) locate much further from visual features. In contrast to centers of visual features (●), text features of different classes overlap each other, showing much smaller inter-class distances.

## A.2 IMPLEMENTATION DETAILS

### A.2.1 NETWORK ARCHITECTURE

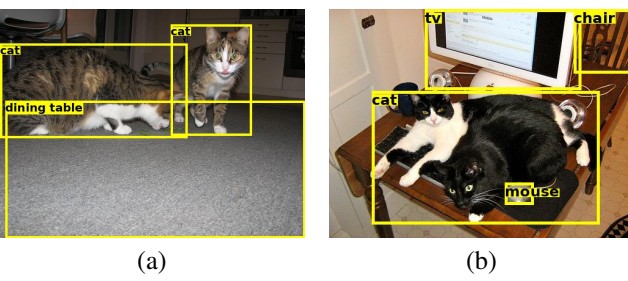

(a)                (b)

Figure A4: Failure cases of our method on COCO. In the image (a), the floor is detected as a dining table, possibly due to visual similarity. In image (b), a single box covers both the two cuddling cats, likely caused by very little visual separation between the two.

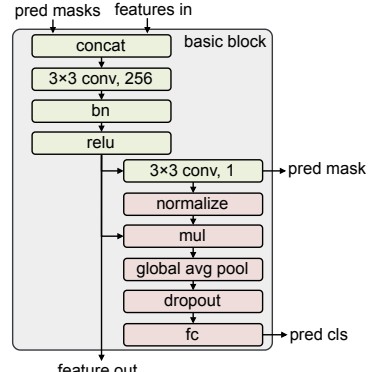

Figure A5: Basic building block for our RCNN networks.

Fig. A5 illustrates the network building block used in our convolution neural networks for classification, localization, and instance segmentation branches. Sigmoid cross entropy loss and dice loss (Sudre et al., 2017) are applied on the mask output at each layer in the localization and segmentation branch. The red blocks are only used in the classification branch. For the classification branch, the mask prediction serves as spatial attention and receives no supervision, similar to the squeeze-and-excitation block in SENet (Hu et al., 2018) but at spatial dimensions. The input to both classification and localization branches is the similarity map between proposal features and prototypes, while the input to the instance segmentation branch for LVIS is the concatenation of similarity maps and multi-scale ViT features. For models trained with COCO, we use 3 blocks for classification and 5 blocks for localization. For LVIS, we use 5 blocks for all branches. The `3×3 conv, 256` denotes a convolution layer with kernel size 3 and 256 output channels. We use $T = 128$ in the class rearrange module for both COCO and LVIS.

### A.2.2 OPERATIONS

**Proposal Expansion.** Eq. 7 explains the proposal expansion block in Fig. 4. $(c_w, c_h, w, h)$ and $(c_w^{\text{exp}}, c_h^{\text{exp}}, w^{\text{exp}}, h^{\text{exp}})$ represent the original the expanded proposal.

$$m = \min(0.4w, 0.4h)$$
$$(w^{\text{exp}}, h^{\text{exp}}) = (w, h) + m \tag{7}$$
$$(c_w^{\text{exp}}, c_h^{\text{exp}}) = (c_w, c_h)$$

**Box2Heatmap.** Eq. 8 explains the Box2Heatmap block in Fig. 4. $\mathbf{H} \in \mathbb{R}^{H \times W}$ represents the output heatmap. $(x_0, y_0)$ and $(x_1, y_1)$ denotes the coordinates of the left top and right bottom point of the original proposal, where $(x_0, y_0) = (c_w, c_h) - (w, h)/2$ and $(x_1, y_1) = (c_w, c_h) + (w, h)/2$ .

$$\mathbf{H}_{ji} = \begin{cases} 1 & (x_0^{\text{exp}} + \frac{i}{W} w^{\text{exp}}, y_0^{\text{exp}} + \frac{j}{H} h^{\text{exp}}) \in [x_0, x_1] \times [y_0, y_1] \\ 0 & \text{otherwise} \end{cases} \tag{8}$$

**Clustering.** The clustering algorithm used in building prototypes is initially proposed by SwAV (Caron et al., 2020) and based on optimal transport (Villani et al., 2009). The optimal transport problem is an optimization problem that aims to find an optimal mapping $\gamma^* \in \mathbb{R}_+^{n \times m}$ from a source distribution $\mathbf{a} = [a_1, ..., a_n], \sum_{i=1}^n a_i = 1$ to target distribution $\mathbf{b} = [b_1, ..., b_n], \sum_{i=1}^m b_i = 1$, which minimizes the overall transport cost. Let $\mathbf{M} \in \mathbb{R}^{n \times m}$ denote the cost matrix, where $\mathbf{M}_{ij}$ denote the cost of moving mass from $a_i$ to $b_j$. This problem is formulated in Eq. 9.

$$\gamma^* = \arg \min_{\gamma \in \mathbb{R}_+^{n \times m}} \sum_{i,j} \gamma_{ij} \mathbf{M}_{ij}$$
$$\text{s.t. } \gamma \mathbf{1} = \mathbf{a}; \gamma^T \mathbf{1} = \mathbf{b}; \gamma \geq 0 \tag{9}$$

The clustering procedure consists of the iterations of two steps. First, an optimal transport $\gamma^*$ is computed between a set of centroids $\mathbf{C} \in \mathbb{R}^{c \times d}$ and instance-level prototypes $\mathbf{Q} \in \mathbb{R}^{q \times d}$, where $c$ represents the number of centroids and $q$ represents the number of instance-level prototypes for a given class in a mini-batch. Note that we apply data augmentation to generate more instance-level prototypes from a limited set of support images. The negative dot similarity $\mathbf{M} = -\mathbf{C}\mathbf{Q}^\top$ is used as the cost matrix, and $\mathbf{a}, \mathbf{b}$ are set as uniform. The solution $\gamma^*$ is estimated from the sinkhorn knopp algorithm (Cuturi, 2013). Second, a momentum update is made to the centroids with $\mathbf{C} = (1 - \beta)\mathbf{C} + \beta\gamma^*\mathbf{Q}$, where $\beta$ is the momentum parameter. Clustering is applied on instance prototypes per class. We use $c = 10$ and $\beta = 0.002$ for all experiments and compute the average of the 10 centroids as the prototype for each class. For background prototypes, we directly use centroids without averaging. This optimal-transport-based online clustering procedure is commonly used in prototype learning (Zhou et al., 2022a; Wang et al., 2022). In hindsight, we also try directly setting class-level prototypes as the mean of instance prototypes for each class without augmentation. This yields almost identical results on open-vocabulary experiments for both COCO and LVIS with ViT-L/14, other settings are not tested. This suggests that the details of the clustering procedure may not be a critical element within our method.

### A.3 VISUALIZATION DETAILS

**Demonstration on YCB Objects.** Fig. 1 shows the detection results of DE-ViT on YCB objects, a standard set of objects widely used in robotic manipulation benchmark (Calli et al., 2015). There are misclassifications and inaccurate boxes, *e.g.*, the white skillet is mistaken as a can, all round-shape fruits are recognized as orange, while the red one is clearly an apple. However, we believe the overall result is encouraging. The specification of YCB objects at the time this paper is written includes 72 categories. We use a total of 33 by selecting and merging certain categories. The categories in use are `apple, ball, banana, bowl, brick, can, cheez-it, chips, clamp, cleanser bottle, coffee jar, comet pine, cups, drill, glass, lego, lemon, marker, mug, mustard, orange, peach, pear, peg-hole, pitcher, plate, screwdriver, skillet, spray bottle, sugar box, toy airplane box, utensil, wood blocks jar`. The source image in Fig. 1 is taken from the banner picture of ycbbenchmarks.com. For each category, we use google image search to collect a few sample images. Fewer than four images on average are gathered per category. We annotate the corresponding objects by instance masks

in each image using the software provided by SimpleClick (Liu et al., 2022). Similar to SAM, SimpleClick generates instance masks automatically from user clicks, which significantly simplifies and accelerates the annotation procedure. Our annotator feedback indicates that annotating masks with SimpleClick is even easier and more accurate than drawing bounding boxes. An NVIDIA 3060 GPU is used for SimpleClick software. Class prototypes for YCB objects are built from the annotated example images. During DE-ViT inference, we replace prototypes of LVIS categories with those of YCB objects in order to detect these new categories. During postprocessing, We apply class-agnostic NMS and filter small bounding boxes. The full code and data for this demonstration will be released in our repository.

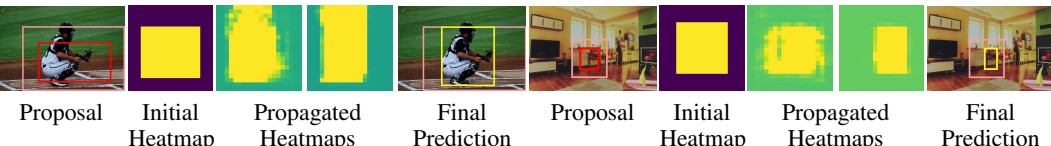

| Proposal | Initial Heatmap | Propagated Heatmaps | Final Prediction | Proposal | Initial Heatmap | Propagated Heatmaps | Final Prediction |

Figure A6: Examples of heatmaps in region propagation. The proposal, expanded proposal, and final prediction box are colored in red, pink, and yellow, respectively. The initial heatmap encodes the spatial relationship between proposals and expanded proposals in binary masks. Propagated heatmaps are softmax predictions sampled from the binary segmentation network in our region propagation architecture.

**Region Propagation.** Fig. A6 visualizes the heatmap changes of sampled detection results during region propagation. A clear mapping can be found between the detection bounding boxes and the corresponding salient regions within the expanded proposal. The softmax predictions are normalized by dividing mean values in order to enhance the visual quality.

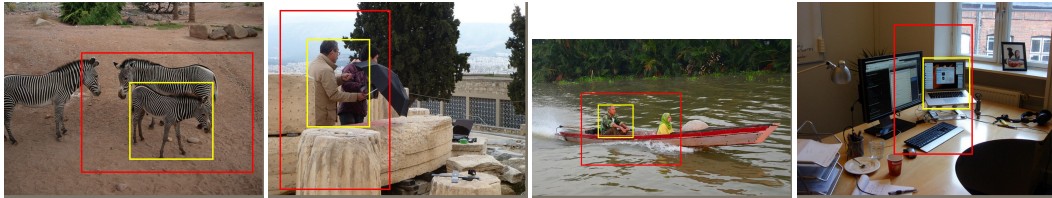

Figure A7: Successful cases of our propagation-based localization under over-expanded proposals. The expanded proposals and final prediction boxes are colored in red and yellow, correspondingly. It can be seen that propagation generally prefers central objects, but can locate an object accurately even if the object is not located at the center of the proposal.

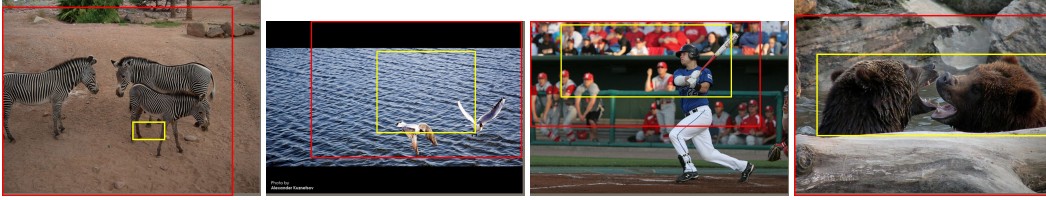

Figure A8: Failure cases of our propagation-based localization under over-expanded proposals. The propagation either encompasses both objects, or produces erratic boxes. But all failure cases happen under inferior proposals, where the proposals are already poorly located and do not cover any object, or cover multiple objects before expansion.

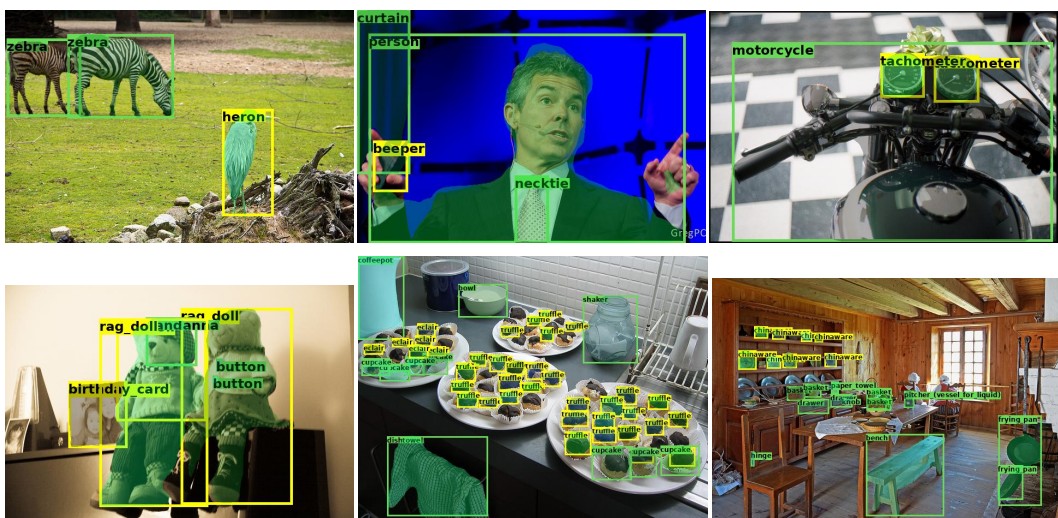

Figure A9: Qualitative visualization of our method DE-ViT on LVIS. Boxes of base and novel classes are colored in green and yellow, correspondingly.

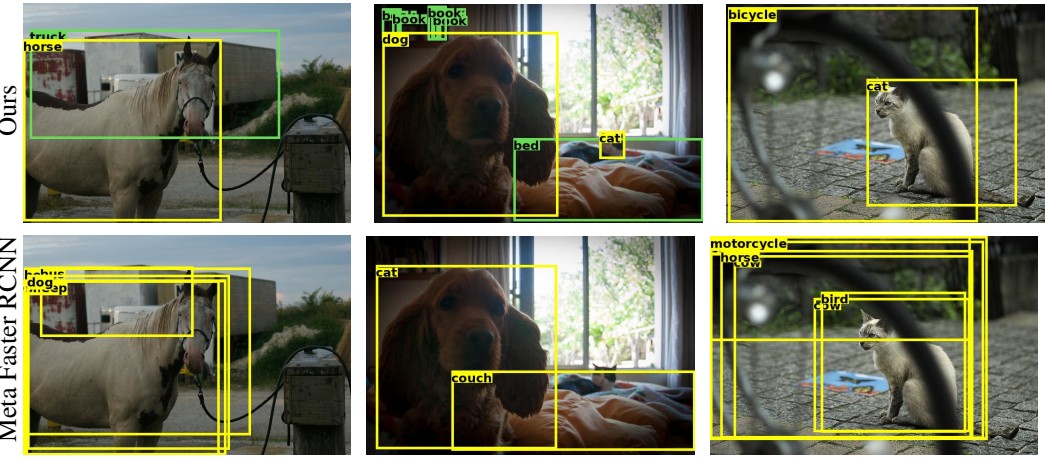

Figure A10: Qualitative comparison between our method DE-ViT with few-shot detector Meta Faster RCNN (Han et al., 2022a) in COCO. DE-ViT detects more novel objects while having much fewer false positives. Note that Meta Faster RCNN can only detect novel objects after finetuning over novel classes, while our method can detect both base and novel classes without any finetuning. Boxes of base and novel classes are colored in green and yellow, correspondingly.

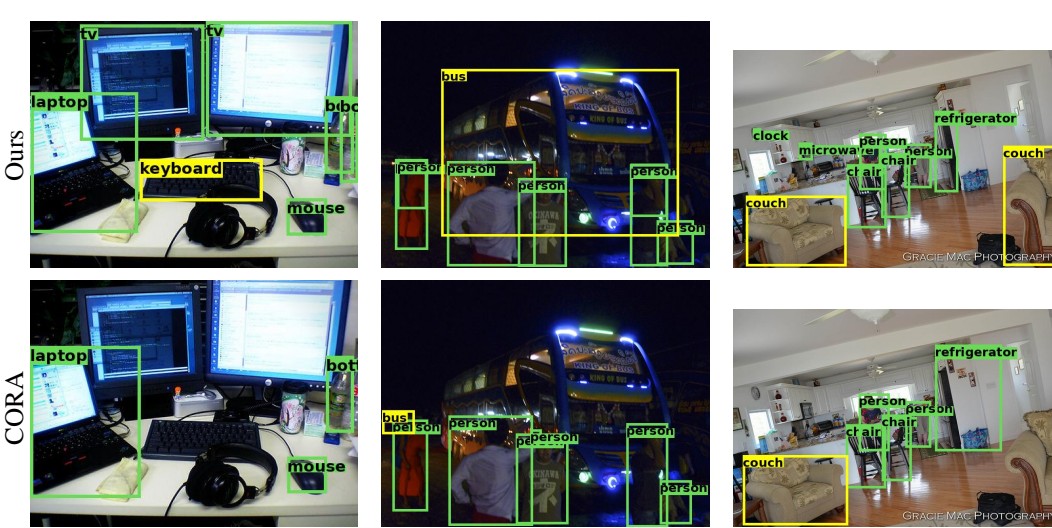

Figure A11: Qualitative comparison between our method DE-ViT with open-vocabulary detector CORA (Wu et al., 2023) in COCO. Boxes of base and novel classes are colored in green and yellow, correspondingly.

