# OpenReview forum: "Detect Every Thing with Few Examples"
_ICLR.cc/2024/Conference — Submitted to ICLR 2024_

### Official Review · Reviewer_Vm41 · 2023-10-30

**Soundness:** 2 fair
**Presentation:** 3 good
**Contribution:** 2 fair
**Rating:** 5
**Confidence:** 5

**Summary:**

This paper proposes to use vision only backbone to represent categories as visual prototype to unify few-shot object detection and open vocabulary detection in cases of input and output. They propose DE-ViT, a combined approach that combine both DINO-v2 and RCNN together. They evaluate DE-ViT on open-vocabulary, few-shot, and one-shot object detection benchmark with COCO and LVIS, which achieves the new state-of-the-art results.

**Strengths:**

- The overall writing is good and easy to follow.

- The proposed method achieves significant improvements on few-shot detection benchmarks. It also achieves strong performance on COCO/LVIS open vocabulary benchmark.

- The proposed refined localization architecture is interesting.

**Weaknesses:**

- The idea of learning prototype to compare is not novel in few-shot learning (detection). In this work, authors adopt more recent strong DINO-v2 to extra visual features to enhance visual entity's comparison.

- The idea is similar to OV-DETR since both uses the bypassing per-class inference procedure.

- I double that the proposed approach may not be a real open vocabulary setting, since the authors assume the novel classes of COCO can be accessed during the inference. This means the vocabulary is still limited and pre-defined. For example, the proposed model is hard to extend into zero-shot setting where the novel prototypes are not available.

- The visual novel objects are sampled from COCO or LVIS, which also may lead to data leakage. What about the classes sampled from ImageNet-21k to build prototypes?

- The model needs extra Region Proposal Network along with backbone. What about the shared frozen DINOv2-ViT backbone design such as works [1][2]?

- Missing parameters and GFLops analysis.

- Presentation: What are the meanings of frozen or unfrozen? Better add the notations in the figure-2.


[1]. Convolutions Die Hard: Open-Vocabulary Segmentation with Single Frozen Convolutional CLIP, NeurIPS-2023

[2]. F-VLM: Open-Vocabulary Object Detection upon Frozen Vision and Language Models, ICLR-2023

**Questions:**

See the weakness part. I rate the current draft as boardline.

---

> ### Author Response · Authors · 2023-11-19
> **Response to reviewer 4 (W1-W2)**
>
> We are greatly thankful to the reviewer for the comments. We hope the following reply resolve the concerns raised in the review.
>
> **(W1, novelty) "The idea of learning prototype to compare is not novel in few-shot learning (detection). In this work, authors adopt more recent strong DINO-v2 to extra visual features to enhance visual entity's comparison."**
>
> We conducted an ablation study (See Section 4.3) to demonstrate that a naive ensemble of DINOv2 and prototype-based few-shot detector leads to poor few-shot performance, and every proposed component is important for the performance of our method. Therefore, combining DINOv2 and prototype-based few-shot detectors is not trivial. Our contributions do not rely on prototype learning and are not simply combining DINOv2 with existing methods.
>
> | Conventional Prototypes Head (baseline) | Prototype Projection | Background Tokens | Class Reorder and Resample | COCO nAP50 | COCO nAP75 |
> | --- | --- | --- | --- | --- | --- |
> | ✓ |  |  |  | 4.5 | 2.2 |
> |  | ✓ |  |  | 26.2 | 9.7 |
> |  | ✓ | ✓ |  | 38.4 | 23 |
> |  | ✓ | ✓ | ✓ | **39.5**| **24.1** |
>
>
> | Conventional  REG Head (baseline) | Expanded Proposal | Region Propagation | COCO nAP50 | COCO nAP75 |
> | --- | --- | --- | --- | --- |
> | ✓ |  |  | 37.7 | 14.6 |
> | ✓ | ✓ |  | 35.6 | 12.5 |
> |  | ✓ | ✓ | **39.5** | **24.1**  |
>
>
> Existing few-shot detection approaches can be broadly classified into finetuning-based [3-6, 9-14] and meta-learning-based strategies [1-2]. Finetuning-based methods demand redundant multi-stage training procedures [7, 8]. Meta-learning methods avoid finetuning by online adaptation, but exhibit inferior accuracy [15].
>
> Contrary to existing work, we are the first to build a few-shot detector on top of DINOv2. Our method transforms the multi-class classification into multiple binary classifications, so a binary classifier can be trained and used for all classes without any finetuning. Our method localizes objects with a novel propagation-based mechanism that we developed. The new mechanism is based on computing similarity maps between features and prototypes.
>
> We revised the introduction section to highlight our technical contributions.
>
>
> [1]: Few-shot object detection via feature reweighting, ICCV 2019
>
> [2]: Meta r-cnn: Towards general solver for instance-level low-shot learning, ICCV 2019
>
> [3]: Frustratingly Simple Few-Shot Object Detection, ICML 2020
>
> [4]: Query adaptive few-shot object detection with heterogeneous graph convolutional networks, ICCV 2021
>
> [5]: Generalized few-shot object detection without forgetting, CVPR 2021
>
> [6]: Fsce: Few-shot object detection via contrastive proposal encoding, CVPR 2021
>
> [7]: Semantic-aligned fusion transformer for one-shot object detection, CVPR 2022
>
> [8]: Balanced and hierarchical relation learning for one-shot object detection, CVPR 2022
>
> [9]: Label, verify, correct: A simple few shot object detection method, CVPR 2022
>
> [10]: Few-shot object detection and viewpoint estimation for objects in the wild, TPAMI 2022
>
> [11]: Meta faster r-cnn: Towards accurate few-shot object detection with attentive feature alignment, AAAI 2022
>
> [12]: Few-shot object detection with fully cross-transformer, CVPR 2022
>
> [13]: NIFF: Alleviating Forgetting in Generalized Few-Shot Object Detection via Neural Instance Feature Forging, CVPR 2023
>
> [14]: DiGeo: Discriminative Geometry-Aware Learning for Generalized Few-Shot Object Detection, CVPR 2023
>
> [15]: Few-shot object detection: a comprehensive survey, IEEE Transaction on Neural Networks and Learning Systems, 2023
>
>
> **(W2, comparision with OV-DETR) "The idea is similar to OV-DETR since both uses the bypassing per-class inference procedure."**
>
> Our idea is not just bypassing per-class inference but also training a binary classifier that can be used for all classes without any finetuning, and a novel propagation-based localization mechanism. On the contrary, OV-DETR [1] does not bypass the per-class inference procedure [2, 3].
>
>
> The inefficiency of OV-DETR is also pointed out by CORA [2] as *“Conditional Matching in OV-DETR  also proposes to condition the queries on the text embedding for class-aware regression. But it suffers from repetitive per-class inference.”*  and Prompt-OVD [3] as *“However, the main limitation of OV-DETR is the need for multiple forward passes in the decoder due to its large number of object queries, which results in slow inference speeds that may not be suitable for real-world use cases."*, and *"OV-DETR needs linearly increasing number of object queries with respect to the number of classes.”*.
>
>
> We updated Section 3.1 at Page 3 of the paper to clarify the procedure of bypassing per-class inference.
>
> Reference:
>
> [1]: Open-vocabulary detr with conditional matching, ECCV 2022
>
> [2]: CORA: Adapting CLIP for Open-Vocabulary Detection with Region Prompting and Anchor Pre-Matching, CVPR 2023
>
> [3]: Prompt-Guided Transformers for End-to-End Open-Vocabulary Object Detection, Arxiv 2023

---

> ### Author Response · Authors · 2023-11-19
> **Response to reviewer 4 (W3-W5)**
>
> **(W3, open vocabulary setting) "I double that the proposed approach may not be a real open vocabulary setting, since the authors assume the novel classes of COCO can be accessed during the inference. This means the vocabulary is still limited and pre-defined. For example, the proposed model is hard to extend into zero-shot setting where the novel prototypes are not available."**
>
> We agree with the reviewer that our proposed method is not an open-vocabulary detector. We revised the entire introduction and experiment section to clarify the  setting and comparison towards open-vocabulary detectors.
>
> Our original intent is to use this comparison to indicate the significant improvement we achieved on few-shot object detection. Because the detection accuracy of existing few-shot methods fall behind that of open-vocabulary detectors, especially on challenging datasets such as LVIS (the open-vocabulary SoTA on LVIS has 31.2 box APr, while the few-shot one only has 16.6 box APr). Our proposed method DE-ViT outperforms the SoTAs on both few-shot and open-vocabulary (33.6 box APr), which has not been achieved before by existing few-shot work.
>
>
> **(W4, data leakage) "The visual novel objects are sampled from COCO or LVIS, which also may lead to data leakage. What about the classes sampled from ImageNet-21k to build prototypes?"**
>
> Sampling base and novel objects from the same datasets is the standard evaluation procedure for few-shot object detection [1].  Regarding the concern for data leakage, all annotations for novel categories are removed during training. So even if the network has seen the images with novel objects inside, the novel objects will be treated as background. There is no data leakage.  In fact, this makes it even more difficult for detecting novel objects that are present in the training set.
>
> Reference:
>
> [1]: Few-shot object detection: a comprehensive survey, IEEE Transaction on Neural Networks and Learning Systems, 2023
>
>
> **(W5, detector over frozen backbone) "The model needs extra Region Proposal Network along with backbone. What about the shared frozen DINOv2-ViT backbone design such as works [1][2]?"**
>
>
> Though both using a frozen backbone, our proposed method DE-ViT delivers more accurate detection results than F-VLM [2] on COCO 2017 (39.5 AP50, 24.1 AP75 vs. F-VLM 28.0 AP50, AP75 not reported), with our propagation-based localization mechanism. Our region proposal network involves small computation cost.
>
>
>
> | Conventional  REG Head (baseline) | Expanded Proposal | Region Propagation | COCO nAP50 | COCO nAP75 |
> | --- | --- | --- | --- | --- |
> | ✓ |  |  | 37.7 | 14.6 |
> | ✓ | ✓ |  | 35.6 | 12.5 |
> |  | ✓ | ✓ | **39.5** | **24.1**  |
>
>
> In Table 8, our proposed method DE-ViT achieves 39.5 AP50 and 24.1 AP75 on COCO with our propagation-based localization. However, when replacing propagation architecture with a conventional regression head, DE-ViT (w contentional regression) only achieves 37.7(-1.8) AP50 and 14.6(-9.5) AP75. The massive drop of AP75 shows the effectiveness of our propagation localization on the frozen DINOv2 backbone. F-VLM [2] trains detectors on frozen backbone using conventional FPN detector head, and has 28.0 AP50. F-VLM does not report AP75.
>
>
>
> Our region proposal network involves a small computation cost. In our model for LVIS, RPN has only **27M** parameters, and our total parameters for ViT-L backbone is 350M. While the F-VLM contains a total parameters of 450M without additional RPN, where the backbone (RN50x64) has 420M parameters. In our model for COCO, RPN has only **8M** parameters.
>
>
> |  | Backbone | Total Params | Trained Params |  APr |
> | --- | --- | --- | --- | --- |
> | F-VLM [2] | RN50x64 | 445M  (backbone 420M) | 25M |  32.8 |
> | DE-ViT (Ours) | ViT-L | **350M** (backbone 320M, RPN 27M) | **23M** | **34.3** |
>
>
>
> We added the mentioned papers [1, 2] into the reference of our paper and incorporated this answer into the efficiency analysis at Section A.1.3 .
>
> Reference:
>
> [1]. Convolutions Die Hard: Open-Vocabulary Segmentation with Single Frozen Convolutional CLIP, NeurIPS 2023
>
> [2]. F-VLM: Open-Vocabulary Object Detection upon Frozen Vision and Language Models, ICLR 2023

---

> ### Author Response · Authors · 2023-11-19
> **Response to reviewer 4 (W6-W7)**
>
> **(W6, parameter analysis) "Missing parameters and GFLops analysis."**
>
> We added the parameters and training epochs comparison for recent detectors on LVIS in Table 6.
>
>
> Compared to the recent detectors trained on LVIS, our method only has 23M trainable parameters, and is trained orders of magnitude faster than F-VLM and OWL-ViT. We also provide inference time comparison in Tables 5, A2, and A3.
>
>
> |  | Backbone | Total Params | Trained Params | Training Epochs | APr |
> | --- | --- | --- | --- | --- | --- |
> | OWL-ViT [3] | ViT-L | 433M | 433M | 1800 | 31.2 |
> | F-VLM [2] | RN50x64 | 445M | 25M | 118 | 32.8 |
> | DE-ViT (Ours) | ViT-L | **350M** | **23M** | **14.4** | **34.3** |
>
>
> | Few-shot Methods | Backbone | Total Params | Trained Params | Inference Time (Sec / Img) | nAP50 |
> | --- | --- | --- | --- | --- | --- |
> | Meta Faster RCNN [4]  | RN101 | 82M | 82M | 0.61 | 31.8 |
> | Cross Transformer [5] | Custom | 24M | 24M | 3 | 35.8 |
> | DE-ViT (Ours) | ViT-S | 70M | **23M** | **0.25** | **42.7** |
>
> | Method | COCO nAP50 | Secs/Img |
> | --- | --- | --- |
> | CORA [6] | 41.7 | 0.5 |
> | DE-ViT (Ours)  | **50** | **0.33** |
>
> | Method | LVIS box APr | Secs/Img |
> | --- | --- | --- |
> | OWL-ViT [3] | 31.2 | **0.42** |
> | DE-ViT (Ours)  | **32.6** | 0.5 |
>
>
>
> For few-shot detection, our proposed method DE-ViT has the smallest inference time of 0.25s/img, compared to 0.61s/img of Meta Faster RCNN and 3s/img of CrossTransformer, while having better performance. DE-ViT also has the smallest trainable parameters. For open-vocabulary detection, DE-ViT is faster (0.33s/img vs 0.5s/img) and more accurate (50 AP50 vs 41.7 AP50) than CORA , the SoTA on COCO. DE-ViT is more accurate (32.6 box APr vs 31.2 box APr) and slightly slower (0.5 s/img vs 0.42 s/img) than OWL-ViT. While our method DE-ViT uses PyTorch, OWL-ViT uses JAX [1]. JAX is commonly regarded as faster in general than PyTorch. Inference time of all methods is measured using  the same machine.
>
> Reference:
>
> [1]: "Composable transformations of Python+NumPy programs: differentiate, vectorize, JIT to GPU/TPU, and more" https://github.com/google/jax
>
> [2]: F-VLM: Open-Vocabulary Object Detection upon Frozen Vision and Language Models, ICLR 2023
>
> [3]: OWL-ViT: Simple open-vocabulary object detection, ECCV 2022
>
> [4]: Meta faster r-cnn: Towards accurate few-shot object detection with attentive feature alignment, AAAI 2022
>
> [5]: Few-shot object detection with fully cross-transformer, CVPR 2022
>
> [6]: CORA: Adapting CLIP for Open-Vocabulary Detection with Region Prompting and Anchor Pre-Matching, CVPR 2023
>
>
> **(W7, missing notation) "Presentation: What are the meanings of frozen or unfrozen? Better add the notations in the figure-2."**
>
> We thank the reviewer for pointing this out. We updated Figure 2 to add a  description for the symbol ❄, which denotes frozen parameters. We removed the symbol ♨, which was used to denote trainable parameters.

---

### Official Review · Reviewer_bhpM · 2023-11-01

**Soundness:** 3 good
**Presentation:** 3 good
**Contribution:** 3 good
**Rating:** 6
**Confidence:** 4

**Summary:**

In this paper, a new method is introduced for few-shot and open-vocabulary object detection. Using DINOv2, the method takes a small set of examples to create main patterns, called prototypes. With these prototypes, the authors describe a way to train a network that can tell apart one item from others, based on the proposals generated by the RPN. They also introduce a technique to improve the initial box predictions. Through detailed tests, the paper shows that their method works better than other current methods. The paper also includes a study that looks closely at how each part of their method contributes to its overall performance.

**Strengths:**

1. The utilization of DINOv2 embeddings to construct prototype for different classes is interesting.
2. The introduction of the propagation procedure offers a novel approach to the problem.
3. The method exhibits exemplary performance, surpassing preceding methodologies across diverse datasets and tasks.
4. The manuscript provides an comprehensive ablation study evaluated the contribution from different modules.

**Weaknesses:**

The writing of Section 3.1 could be enhanced. Some portions might be presented in a more reader-friendly manner.

**Questions:**

1. Would it be possible to incorporate notations such as `f` and `h` into Figure 3 for easier referencing?
2. Should the notation be "[C]\\c_k" rather than "[C]/c_k"?
3. The upsampling process described in Eq.3 is somewhat unclear. Could you first elucidate the underlying objective or insight you aim to convey? The rationale behind upsampling on the channel dimension is not immediately evident. If the intent is to balance the number of prototypes and base categories, might a re-weighting approach similar to the focal loss be more appropriate?
4. During the propagation procedure, what transpires if the expanded box encompasses another object of the same category?
5. Could you provide a more detailed explanation of Eq. 4, particularly concerning the `w` and `h` components? An illustrative figure might enhance comprehension.
6. Clarification on terms such as "bAP" and "Split-n" would be beneficial.
7. Are there prior methods that employ DINO/DINOv2 embeddings to craft representations for detection/re-identification? If so, what's the difference between their methods and yours?
8. For the definition of h, it might be better to use "h=\\bar{f}\\dot p" directly to create better reading experience.
9. What are the limitations and failure cases of this work?

---

> ### Author Response · Authors · 2023-11-19
> **Response to reviewer 3 (W1-4, 8)**
>
> We are greatly thankful to the reviewer for the comments and acknowledgement of our work. We hope the following reply resolves the raised concerns.
>
> **(W1 W2 W8, fix notation) "1. Would it be possible to incorporate notations such as f and h into Figure 3 for easier referencing?
> 2. Should the notation be "[C]\c_k" rather than "[C]/c_k"?
> 3. For the definition of h, it might be better to use "h=\bar{f}\dot p" directly to create better reading experience."**
>
> We thank the reviewer for pointing out the notation issue. We updated Figure 3 to include f and h notations as requested. We fixed all equations and figures to change $[C]/c_k$ to $[C]\backslash c_k$. We also rewrote the definition of h to use dot product.
>
>
> **(W3, Eq3) "The upsampling process described in Eq.3 is somewhat unclear. Could you first elucidate the underlying objective or insight you aim to convey? The rationale behind upsampling on the channel dimension is not immediately evident. If the intent is to balance the number of prototypes and base categories, might a re-weighting approach similar to the focal loss be more appropriate?"**
>
>
> Eq.3 sorts and resamples the input similarity map from shape $H\times W\times (C-1)$ to $H\times W\times T$, where $C$ is the number of classes and $T$ is a fixed hyper-parameter.
>
>
> In details, since the number of classes $C$ is different per dataset, the shape of the similarity map is also different, which is problematic because the similarity map is used as the input to the binary classification network, which is the same for all datasets. Therefore, in order to standardize the input size, we resample them to $H\times W\times T$. The intent is not to balance the number of prototypes and base categories.
>
> We updated the paper to incorporate this answer in Section 3.1.
>
>
> **(W4 - propagation procedure) "During the propagation procedure, what transpires if the expanded box encompasses another object of the same category?"**
>
> We added a visual analysis of the propagation localization from expanded proposals covering same-class objects in the Section A.1.1 at Page 14. We will use the term over-expanded proposals to denote the scenario mentioned by the reviewer.
>
>
> | Score Threshold | Average IoU of refined boxes with groundtruth from over-expanded Proposals | Average IoU of refined boxes with groundtruth for other proposals |
> | --- | --- | --- |
> | 0 | 0.49 | 0.59 |
> | 0.1 | 0.56 | 0.69 |
> | 0.2 | 0.58 | 0.72 |
> | 0.3 | 0.60 | 0.74 |
> | 0.4 | 0.62 | 0.75 |
> | 0.5 | 0.63 | 0.77 |
> | 0.6 | 0.65 | 0.78 |
> | 0.7 | 0.68 | 0.80 |
> | 0.8 | 0.72 | 0.83 |
> | 0.9 | 0.76 | 0.86 |
> | 0.95 | 0.82 | 0.87 |
>
>
>
>
> As shown in Figure A1, over-expanded proposals generally degrade localization accuracy, as their final predicted boxes have smaller IOU towards the ground truth. But the degradation is far from total, e.g., they still produce boxes whose IOU > 0.7 on average, under the score threshold of 0.85,  which we use to generate qualitative visualizations. We also observe that the over-expanded proposals occupy around 7% of all proposals in our model for COCO, and only half of them appear in the final prediction (after NMS and score filtering). This means even if an over-expanded proposal predicts an inaccurate box, its impact is softened by filtering of NMS and score thresholding.
>
> In terms of actual behavior, we listed success and failure cases of over-expanded proposals in Figure A7 and A8. In failure cases, propagation either encompasses both objects, or produces erratic boxes. But all failure cases happen under inferior proposals, where the proposals are already poorly located and do not cover any object, or cover multiple objects before expansion. In success cases, propagation generally prefers central objects, but can locate an object accurately regardless of the proposal quality. This means our propagation localization does not fully rely on the proposal quality.

---

> ### Author Response · Authors · 2023-11-19
> **Response to reviewer 3 (W5, W6)**
>
> **(W5 - Eq4) "Could you provide a more detailed explanation of Eq. 4, particularly concerning the w and h components? An illustrative figure might enhance comprehension."**
>
> The first line Eq.4 computes the expected position of the heatmap as the bounding box center, the second line computes a weighted average of row sum of the heatmap as the bounding box width, and column sum as the height.
>
>
> In more details, consider a toy example of converting a binary mask into a bounding box, a reasonable approach is to compute the mask center as the bounding box center, and pick the maximum row and column sum as width and height, as illustrated in Figure **[X]**. Motivated by this, our method estimates the center by computing the expected position under the spatial distribution $softmax(\textbf{g})$, where $\textbf{g}$ denotes the heatmap values. To emulate the row / column sum, we compute  $\sum_{j=1}^W  \sigma(\textbf{g})_{ij}$, which sums up the sigmoid activation along the i-th row. Instead of picking the maximum, we aggregate all row sums in terms of magnitude. The rationale of this aggregation is that a larger estimation is more likely to over-cover the entire object and a small estimation may produce a box that is too tight. The aggregation is done by sorting the estimation and then weighted averaging, which explains the use of order statistics notation $(i)$ and $(j)$.
>
> We updated the paper to incorporate this answer in Section 3.2 and add an illustrative example in Figure 5.
>
>
> **(W6 -evaluation metrics) "Clarification on terms such as "bAP" and "Split-n" would be beneficial."**
>
> "bAP" denotes mAP on base classes. For "Split-n", one-shot methods divide 80 classes of COCO2017 into four even partitions, and alternatively takes three partitions as base classes and one partition as novel classes. There are 4 base/novel splits in total, named as Split-1/2/3/4.
>
>
>
> We clarify the evaluation metrics used in the papers. We updated Paragraph 2 in Section 3 for a more comprehensive description of evaluation metrics.
>
> - Few-shot
>     - bAP: mAP on base classes
>     - nAP, nAP50, nAP75: mAP, AP50, AP75 on novel classes
> - One-shot
>     - Seen -> Split-1/2/3/4: AP50 on base classes, for each split
>     - Unseen -> Split-1/2/3/4: AP50 on novel classes, for each split
>     - Avg: AP50 averaged among all splits, for base or novel classes
> - Open-vocabulary
>     - COCO
>         - Novel: AP50 on novel classes
>         - Base: AP50 on base classes
>         - All: AP50 on all classes
>     - LVIS
>         - APr: AP on rare categories, which are novel classes
>         - AP: AP on all classes
>         - APc: AP on common categories, which are base classes
>         - APf: AP on frequent categories, which are base classes

---

> ### Author Response · Authors · 2023-11-19
> **Response to reviewer 3 (W7, W8)**
>
> **(W7-prior work that uses DINO/DINOv2) "Are there prior methods that employ DINO/DINOv2 embeddings to craft representations for detection/re-identification? If so, what's the difference between their methods and yours?"**
>
>
> To the best of our knowledge, we are the first to incorporate DINO/DINOv2 into the few-shot object detection task.
>
> OW-DETR [1] proposes to use DINO features in open-world object detection tasks, while our proposed method DE-ViT is used in few-shot object detection. The difference between open-world detection and few-shot detection is that open-world detection marks all novel objects as unknown, while few-shot detection detects novel categories based on the support set. OW-DETR uses the attention map of DINO as pseudo labels for unknown objects, i.e., unlabeled areas with high attention saliency is more likely to have an unknown object. But OW-DETR only detects and does not classify unknown objects into specific categories. Our DE-ViT uses the similarity map between DINOv2 features and prototypes to detect and classify novel objects.
>
> NormCut [2] proposes a graph-cut method on top of DINO features to detect the most salient object in images. However, it can only detect a single object for each image, while our DE-ViT does not have the limitation on number of objects.
>
> Reference:
>
> [1]: Ow-detr: Open-world detection transformer, CVPR 2022
>
> [2]: Self-supervised transformers for unsupervised object discovery using normalized cut, CVPR 2022
>
>
>
> **(W9, limitation) "What are the limitations and failure cases of this work?"**
>
>
>
> One of the limitations is that our current architecture is a mix of ViT and RCNN, while a full transformer network can clearly be more scalable and unlock more possible abilities and integrations to other modalities. Another limitation is that our current model relies on an external region proposal network (RPN). It is possible to train an RPN on top of frozen DINOv2 with some engineering efforts.
>
>
> We added some failure examples on COCO in Figure A4 in Appendix. Image (a) in Figure A4 detects the floor as a dining table, possibly due to visual similarity. Image (b) shows a single box over two cuddling cats, likely caused by very little visual separation between the two.  We discussed limitations of our work in Section 5. We updated the paper to include more qualitative results and failure case visualization in Figure 8, A4, A9, A10, and A11.

---

### Official Review · Reviewer_f8VP · 2023-11-02

**Soundness:** 2 fair
**Presentation:** 3 good
**Contribution:** 2 fair
**Rating:** 5
**Confidence:** 4

**Summary:**

This paper proposed an open-set object detector using vision-only DINOv2 backbone to learn new classes through images rather than language. The paper proposed a new region propagation technique for localization and evaluate the method on open-vocabulary, few-shot and one-shot benchmark. The experimental results show good improvement.

**Strengths:**

1. The paper is well-written and presented.
2. The description of methods is clear.

**Weaknesses:**

1. The method lacks some novelty and doesn't look attractive to me. This is the main problem of the paper. So many parts ensembles together only cost more computational resources.
2. On the few-shot benchmark, the good experiments are more shown on 10 or more shot rather than fewer-shot. There are so many different measures  shown in the experimental results, which are not convincing to me about its real performance.
3. There is no visualization of the experimental results shown in the paper.

**Questions:**

please see the weakness.

---

> ### Author Response · Authors · 2023-11-19
> **Response to reviewer 2 (W1, part1)**
>
> We are greatly thankful to the reviewer for the comments. We hope the following reply resolve the concerns raised in the review.
>
> **(W1, part1, novelty)  "The method lacks some novelty and doesn't look attractive to me. This is the main problem of the paper. "**
>
>
> Existing few-shot detection approaches can be broadly classified into finetuning-based [3-6, 9-14] and meta-learning-based strategies [1-2]. Finetuning-based methods demand redundant multi-stage training procedures [7, 8]. Meta-learning methods avoid finetuning by online adaptation, but exhibit inferior accuracy [15].
>
> Contrary to existing work, we are the first to build a few-shot detector on top of DINOv2. Our method transforms the multi-class classification into multiple binary classifications, so a binary classifier can be trained and used for all classes without any finetuning. Our method localizes objects with a novel propagation-based mechanism that we developed. The new mechanism is based on computing similarity maps between features and prototypes.
>
>
>
> Our method outperforms existing few-shot detection models by a significant margin on COCO (+15 nAP on 10shot, +7.2 nAP on 30shot) and LVIS (+20 box APr) while eliminating the finetuning requirement. We conducted an ablation study (See Section 4.3) to demonstrate that every proposed component is important for the performance of our proposed method.
>
> | Conventional Prototypes Head (baseline) | Prototype Projection | Background Tokens | Class Reorder and Resample | COCO nAP50 | COCO nAP75 |
> | --- | --- | --- | --- | --- | --- |
> | ✓ |  |  |  | 4.5 | 2.2 |
> |  | ✓ |  |  | 26.2 | 9.7 |
> |  | ✓ | ✓ |  | 38.4 | 23 |
> |  | ✓ | ✓ | ✓ | **39.5**| **24.1** |
>
>
> | Conventional  REG Head (baseline) | Expanded Proposal | Region Propagation | COCO nAP50 | COCO nAP75 |
> | --- | --- | --- | --- | --- |
> | ✓ |  |  | 37.7 | 14.6 |
> | ✓ | ✓ |  | 35.6 | 12.5 |
> |  | ✓ | ✓ | **39.5** | **24.1**  |
>
>
>
> We revised the introduction section to highlight our technical contributions.
>
> Reference:
>
> [1]: Few-shot object detection via feature reweighting, ICCV 2019
>
> [2]: Meta r-cnn: Towards general solver for instance-level low-shot learning, ICCV 2019
>
> [3]: Frustratingly Simple Few-Shot Object Detection, ICML 2020
>
> [4]: Query adaptive few-shot object detection with heterogeneous graph convolutional networks, ICCV 2021
>
> [5]: Generalized few-shot object detection without forgetting, CVPR 2021
>
> [6]: Fsce: Few-shot object detection via contrastive proposal encoding, CVPR 2021
>
> [7]: Semantic-aligned fusion transformer for one-shot object detection, CVPR 2022
>
> [8]: Balanced and hierarchical relation learning for one-shot object detection, CVPR 2022
>
> [9]: Label, verify, correct: A simple few shot object detection method, CVPR 2022
>
> [10]: Few-shot object detection and viewpoint estimation for objects in the wild, TPAMI 2022
>
> [11]: Meta faster r-cnn: Towards accurate few-shot object detection with attentive feature alignment, AAAI 2022
>
> [12]: Few-shot object detection with fully cross-transformer, CVPR 2022
>
> [13]: NIFF: Alleviating Forgetting in Generalized Few-Shot Object Detection via Neural Instance Feature Forging, CVPR 2023
>
> [14]: DiGeo: Discriminative Geometry-Aware Learning for Generalized Few-Shot Object Detection, CVPR 2023
>
> [15]: Few-shot object detection: a comprehensive survey, IEEE Transaction on Neural Networks and Learning Systems, 2023

---

> ### Author Response · Authors · 2023-11-19
> **Response to reviewer 2 (W1, part2-3)**
>
> **(W1, part2, computational cost) "So many parts ensembles together only cost more computational resources."**
>
> We compared our proposed method DE-ViT with recent few-shot work (see Table 5). Our proposed DE-ViT has shortest inference time 0.25s/img, compared to 0.61s/img for Meta Faster RCNN and 3s/img for CrossTransformer, while having significantly higher detection accuracy. DE-ViT also has the smallest number of trainable parameters (23M). The inference time of all compared methods is measured using the same machine.
>
>
> It is worth noting that the few-shot detection SOTA LVC [1] requires a total of 18 stages for self-training and pseudo-labeling procedures [2]. A pretrained model for LVC has never been released.  Other recent few-shot works also include multiple pretraining and finetuning stages [3, 4].  Our proposed method DE-ViT can be trained in a single stage and used on novel objects directly without any finetuning.
>
>
>
> | Few-shot methods | Backbone | Total Params | Trained Params | Inference Time (Sec / Img) | nAP50 |
> | --- | --- | --- | --- | --- | --- |
> | Meta Faster RCNN [3] | RN101 | 82M | 82M | 0.61 | 31.8 |
> | Cross Transformer [4] | Custom | 24M | 24M | 3 | 35.8 |
> | DE-ViT (Ours) | ViT-S | 70M | **23M** | **0.25** | **42.7** |
>
>
>
>
> Compared against the recent  detectors trained on the large-scale dataset LVIS, our proposed method DE-ViT has the smallest number of total parameters, and is trained orders of magnitude faster than F-VLM and OWL-ViT.
>
>
> |  | Backbone | Total Params | Trained Params | Training Epochs | APr |
> | --- | --- | --- | --- | --- | --- |
> | OWL-ViT | ViT-L | 433M | 433M | 1800 | 31.2 |
> | F-VLM | RN50x64 | 445M | 25M | 118 | 32.8 |
> | DE-ViT (Ours) | ViT-L | **350M** | **23M** | **14.4** | **34.3** |
>
>
> We updated the paper to include the inference time and parameter size comparison in Section 4.2 at Page 8.
>
> Reference:
>
> [1]: Label, verify, correct: A simple few shot object detection method, CVPR 2022
>
> [2]: Official github repo for "Label, verify, correct: A simple few shot object detection method" https://github.com/prannaykaul/lvc
>
> [3]: Meta faster r-cnn: Towards accurate few-shot object detection with attentive feature alignment, AAAI 2022
>
> [4]: Few-shot object detection with fully cross-transformer, CVPR 2022
>
>
>
> **(W1, part3, parts ensemble) "So many parts ensembles together only cost more computational resources."**
>
>
> We did an ablation study to show that a naive ensemble of DINOv2 and prototype-based few-shot detector leads to poor few-shot performance (See Section 4.3). Therefore, each one of our proposed component is critical for our results.
>
>
> | Conventional Prototypes Head | Prototype Projection | Background Tokens | Class Reorder and Resample | nAP50 | nAP75 |
> | --- | --- | --- | --- | --- | --- |
> | ✓ |  |  |  | 4.5 | 2.2 |
> |  | ✓ |  |  | 26.2 | 9.7 |
> |  | ✓ | ✓ |  | 38.4 | 23 |
> |  | ✓ | ✓ | ✓ | **39.5** | **24.1** |
>
>
> Regarding the concern related to the number of parts, we clarify that our method only has three trainable components, i.e., the Binary Classification Network, the Binary Segmentation Network, and the Spatial Integral Layer, as shown in Figures 2, 3. The number of parts is similar, if not smaller, than those of existing works [1-3].
>
> In comparison, CrossTransformer [1] includes 4 stages of dense feature cross interactions, and each stage is trained separately. Meta Faster RCNN [2] has fours trainable components: Meta-RPN, Meta-Classifier, Base-RPN and Base-Classifier. CORA [3] has 3 components: region classifier, anchor pre-matching, and deformable DETR encoder/decoder.
>
> Reference:
>
> [1]: Meta faster r-cnn: Towards accurate few-shot object detection with attentive feature alignment, AAAI 2022
>
> [2]: Few-shot object detection with fully cross-transformer, CVPR 2022
>
> [3]: CORA: Adapting CLIP for Open-Vocabulary Detection with Region Prompting and Anchor Pre-Matching, CVPR 2023

---

> ### Author Response · Authors · 2023-11-19
> **Response to reviewer 2 (W2, part1)**
>
> **(W2, part1, performance on fewer shots) "On the few-shot benchmark, the good experiments are more shown on 10 or more shot rather than fewer-shot. "**
>
>
> Regarding the results on fewer shots, it is common practice for few-shot detection work to only report 10, 30 shots results on COCO, such as [1-4].  In the revised paper, we included plots of the model performance for fewer shots in Figure 7 and A2 for COCO 2014 and COCO 2017, correspondingly. In Section 4.2, we identified the inflection points for each dataset, after which adding more shots does not help. Our model outperforms existing few-shot SOTA results on all shots.
>
> We updated the experiment section to include the performance analysis with more shots in Section 4.2 and Section A.1.2.
>
> Reference:
>
> [1]: Frustratingly Simple Few-Shot Object Detection, ICML 2020
>
> [2]: Label, verify, correct: A simple few shot object detection method, CVPR 2022
>
> [3]: Few-shot object detection and viewpoint estimation for objects in the wild, TPAMI 2022
>
> [4]: DiGeo: Discriminative Geometry-Aware Learning for Generalized Few-Shot Object Detection, CVPR 2023

---

> ### Author Response · Authors · 2023-11-19
> **Response to reviewer 2 (W2, part2)**
>
> **(W2, part2, evaluation measures and real performance) "There are so many different measures shown in the experimental results, which are not convincing to me about its real performance."**
>
> We clarify the measures used in the paper. In the few-shot/one-shot/open-vocabulary detection tasks, classes are split into base and novel [2-4]. Base classes are seen during training and novel classes are unseen. The performance on novel classes is more important. One-shot detection methods conventionally use the term seen / unseen classes, but it has the same meaning as base / novel classes.  One-shot methods conventionally divide 80 classes of COCO into four even partitions, and alternatively takes three partitions as base classes and one partition as novel classes.  There are 4 base/novel splits in total, named as split-1/2/3/4. Metrics on LVIS are computed separately on bounding boxes (e.g., box AP) or instance segmentation masks (e.g., mask AP) [1].
>
>
>
> List of all measures:
>
> - Few-shot
>     - bAP: mAP on base classes
>     - nAP, nAP50, nAP75: mAP, AP50, AP75 on novel classes
> - One-shot
>     - Seen -> Split-1/2/3/4: AP50 on base classes, for each split
>     - Unseen -> Split-1/2/3/4: AP50 on novel classes, for each split
>     - Avg: AP50 averaged among all splits, for base or novel classes
> - Open-vocabulary
>     - COCO
>         - Novel: AP50 on novel classes
>         - Base: AP50 on base classes
>         - All: AP50 on all classes
>     - LVIS
>         - APr: AP on rare categories, which are novel classes
>         - AP: AP on all classes
>         - APc: AP on common categories, which are base classes
>         - APf: AP on frequent categories, which are base classes
>
>
>
>
> For few-shot detection, our proposed method DE-ViT outperforms existing SoTA significantly on nAP at 10-shot (+15) and 30-shot (+7.2) in COCO, and +20 box APr on LVIS. For one-shot detection, DE-ViT outperforms SoTA on unseen classes by 2.8 AP50. When compared with open-vocabulary detectors, DE-ViT outperforms SoTA by 6.9 AP50 on COCO, and 1.5 mask APr on LVIS. In summary, our proposed method DE-ViT outperforms all existing solutions on detecting objects of novel categories.
>
> | Few-shot Method on COCO | Finetune on Novel | nAP (10-shot) | nAP (30-shot) |
> | --- | --- | --- | --- |
> | Meta Faster R-CNN [5] | ✓ | 12.7 | 16.6 |
> | Cross-Transformer [7]  | ✓ | 17.1 | 21.4 |
> | LVC [6] | ✓ | 19 | 26.8 |
> | DE-ViT (Ours) | ✗ | **34** | **34** |
>
> | Few-shot Method on LVIS | APr | APc | APf | AP |
> | --- | --- | --- | --- | --- |
> | DiGeo [9] | 16.6 | 22.8 | 28 | 24.4 |
> | DE-ViT (Ours) | **33.6** | **30.1** | **30.7** | **30.9** |
>
>
> | One-shot Method on COCO | bAP50 | nAP50 |
> | --- | --- | --- |
> | BHRL [3] | 53.6 | 25.6 |
> | SaFT [8] | 48.3 | 24.9 |
> | DE-ViT (Ours) | **59.6** | **28.45** |
>
> | Open-vocabulary Method | Use Extra Training Set | LVIS mask APr | LVIS box APr | COCO nAP50 |
> | --- | --- | --- | --- | --- |
> | CORA+ [10] | ✓ | - | 28.1 | 43.1 |
> | RO-ViT [11] | ✗ | 32.1 | - | 34 |
> | F-VLM [12] | ✗ | 32.8 | - | 28 |
> | DE-ViT (Ours) | ✗ | **34.3** | **33.6** | **50** |
>
>
>
>
> We updated Section 4 Paragraph 2 to have a more comprehensive description of the measures, and unify the measure names in Tables 1, 2 and 4.
>
> Reference:
>
> [1]: Lvis: A dataset for large vocabulary instance segmentation, CVPR 2019
>
> [2]: Frustratingly Simple Few-Shot Object Detection, ICML 2020
>
> [3]: Balanced and hierarchical relation learning for one-shot object detection, CVPR 2022
>
> [4]: Regionclip: Region-based language-image pretraining, CVPR 2022
>
> [5]: Meta faster r-cnn: Towards accurate few-shot object detection with attentive feature alignment, AAAI 2022
>
> [6]: Label, verify, correct: A simple few shot object detection method, CVPR 2022
>
> [7]: Few-shot object detection with fully cross-transformer, CVPR 2022
>
> [8]: Semantic-aligned fusion transformer for one-shot object detection, CVPR 2022
>
> [9]: DiGeo: Discriminative Geometry-Aware Learning for Generalized Few-Shot Object Detection, CVPR 2023
>
> [10]: CORA: Adapting CLIP for Open-Vocabulary Detection with Region Prompting and Anchor Pre-Matching, CVPR 2023
>
> [11]: Region-aware pretraining for open-vocabulary object detection with vision transformers, CVPR 2023
>
> [12]: F-vlm: Open-vocabulary object detection upon frozen vision and language models, ICLR 2023

---

> ### Author Response · Authors · 2023-11-19
> **Response to reviewer 2 (W3)**
>
> **(W3, visualizations) "There is no visualization of the experimental results shown in the paper."**
>
> In the original submission, we have the visualization on detecting YCB objects with our proposed method DE-ViT in Figure 1. In the revised version, we added Figures 8, A9, A10, and A11 to include qualitative results and comparisons of our method and Meta Faster RCNN [1], CORA [2] on COCO and LVIS datasets.  DE-ViT detects more novel objects while having much fewer false positives, as shown in Figures A10 and A11. Base and novel objects are colored in "green" and "yellow", respectively.
>
> Reference:
>
> [1]: Meta faster r-cnn: Towards accurate few-shot object detection with attentive feature alignment, AAAI 2022
>
> [2]: CORA: Adapting CLIP for Open-Vocabulary Detection with Region Prompting and Anchor Pre-Matching, CVPR 2023

---

### Official Review · Reviewer_Jjnu · 2023-11-05

**Soundness:** 3 good
**Presentation:** 3 good
**Contribution:** 3 good
**Rating:** 6
**Confidence:** 4

**Summary:**

This paper proposes a paradigm for few-shot object detection which does not rely on fine-tuning the base model or top layers for the few-shot classes. The authors compare the paradigm with few-shot detection, open-vocabulary detection, and one-shot detection. The authors propose to use a few samples from novel classes to create prototypes for objects which can be used to detect new objects in images. The proposed paradigm leads to improved performance compared to prior few-shot, one-shot, and open-vocabulary methods.

**Strengths:**

The paper proposes a different paradigm for few-shot object detection which alleviates the need for fine-tuning during the few-shot inference process. The paper also shows that this paradigm can lead to improved performance compared to prior methods.

**Weaknesses:**

The main weaknesses of the paper are based around a lack of clear motivations for some decisions and some un-fair comparisons. In particular, the authors should respond to the following:

1. I believe that the comparisons to standard open-vocabulary object detection methods are unfair because the setting proposed in this paper is quite different - open-vocabulary (or zero-shot object detection) does not assume any knowledge of visual representation of novel classes, however, the proposed approach does. So, placing this method as an open-vocabulary object detection approach is unfair, and frankly, unnecessary. The authors can focus on few-shot/one-shot approaches - this is already a good paper for these settings.

2. In figure 1, which objects come from based classes and which are the few-shot/novel categories? There are different colours used in the image but no legend. This makes it difficult to ascertain the performance of the proposed approach based on this image.

3. If you are selecting the top-K classes in one step, why is there a need to up-sample to T? Can't we just keep using "K" because that is deterministic - just up/down-sample to K?

4. How does the performance change with the number of samples used for prototype creation? From Table 3, it seems that the performance for 30-shot is lower than for 10-shot in some cases. Why? Is there an inflection point where adding more samples stops helping?

5. Do prior one-shot object detection methods also follow the few-shot paradigm of fine-tuning the model? If not, what exactly is the difference between the proposed method and one-shot methods?

6. Suggestion: It might be better to show equation 1 (and other similar things) as dot-products to make it more straight-forward to understand.

**Questions:**

Please see weaknesses above.

---

> ### Author Response · Authors · 2023-11-19
> **Response to reviewer 1 (W1 - W4)**
>
> We are greatly thankful to the reviewer for the comments and acknowledgement of our work. We hope the following reply resolves the raised concerns.
>
> **(W1, open-vocabulary setting) "The comparisons to standard open-vocabulary object detection methods are unfair because the setting proposed in this paper is quite different - open-vocabulary (or zero-shot object detection) does not assume any knowledge of visual representation of novel classes, however, the proposed approach does. So, placing this method as an open-vocabulary object detection approach is unfair, and frankly, unnecessary. The authors can focus on few-shot/one-shot approaches - this is already a good paper for these settings."**
>
> We agree with the reviewer that our proposed method is not an open-vocabulary detector. We revised the entire introduction and experiment sections to focus on few-shot/one-shot settings, clarify the comparison to open-vocabulary detectors. We also revised the method section to clarify the motivations for our technical designs.
>
>
> Our original intent is to use this comparison to indicate the significant improvement we achieved on few-shot object detection. Because the detection accuracy of existing few-shot methods fall behind that of open-vocabulary detectors, especially on challenging datasets such as LVIS (the open-vocabulary SoTA on LVIS has 31.2 box APr, while the few-shot one only has 16.6 box APr).
>
> Our proposed method DE-ViT outperforms the SoTAs on both few-shot and open-vocabulary (33.6 box APr), which has not been achieved before by existing few-shot work.
>
>
>
> **(W2, Figure 1) "In figure 1, which objects come from based classes and which are the few-shot/novel categories? There are different colours used in the image but no legend. This makes it difficult to ascertain the performance of the proposed approach based on this image."**
>
>
>
> We added legends and colors to differentiate the base and novel categories in Figure 1. Specifically, "green" denotes base classes, and “yellow” denotes novel classes. Our proposed method DE-ViT's performance is demonstrated by detecting novel classes such as: mustard, clamp, cheez-it, coffee jar, peg-hole, toy airplane box, comet pine, wood blocks jar, chips, spray bottle, cleaser bottle.
>
>
> **(W3, K vs T) "If you are selecting the top-K classes in one step, why is there a need to up-sample to T? Can't we just keep using "K" because that is deterministic - just up/down-sample to K?"**
>
> $K$ and $T$ serve different purposes. $K$ is used to select classes, reducing unnecessary computation. $T$ is used to standardize the shape of the similarity map, while preserving information. So they should not be set identical.
>
>
> In details, we select top $K$ classes to estimate class score, so we want $K$ to be as small as possible. In practice, we use $K=3$ or $K=10$ in our experiments. To estimate probability for each selected class, the input is a similarity map $H\times W\times C$ between features and prototypes, where $C$ denotes number of classes. As $C$ is different per dataset, we standardize the input shape to $H\times W\times T$, where $T$ is a hyperparameter and set to 128 in our experiments.
>
>
> We updated the paper to incorporate this answer in Section 3.1 .
>
>
>
>
> **(W4, shots performance) "How does the performance change with the number of samples used for prototype creation? From Table 3, it seems that the performance for 30-shot is lower than for 10-shot in some cases. Why? Is there an inflection point where adding more samples stops helping?"**
>
>
> Performance generally increases with the number of shots. The performance difference between 30-shot (nAP50=52.9) and 10-shot (nAP50=53.0) is smaller than 0.1% and could be interpreted as statistically insignificant. There exists indeed inflection point after which more samples do not help.
>
> | shots | nAP50 (COCO 2014) |
> | --- | --- |
> | 1 | 43.3 |
> | 2 | 50.1 |
> | 3 | 52.5 |
> | 4 | 52.7 |
> | 5 | 52.8 |
> | 6 | 53 |
> | 7 | 52.8 |
> | 8 | 52.9 |
> | 9 | 53 |
> | 10 | 53 |
> | 20 | 52.8 |
> | 30 | 52.9 |
>
> | shots | nAP50 (COCO 2017) |
> | --- | --- |
> | 1 | 22.1 |
> | 2 | 29.9 |
> | 3 | 33.8 |
> | 4 | 36.3 |
> | 5 | 37.3 |
> | 6 | 38.5 |
> | 7 | 39.4 |
> | 8 | 39.8 |
> | 9 | 40.5 |
> | 10 | 40.7 |
> | 15 | 42.3 |
> | 20 | 42.4 |
> | 30 | 43.2 |
> | 40 | 43.5 |
> | 50 | 43.7 |
> | 75 | 43.9 |
> | 100 | 43.9 |
>
>
>
> We ploted the nAP50 with different shots in Figure 7 and Figure A2 for COCO 2014 and COCO 2017, correspondingly. The inflection point is located around 50 to 75 shots for COCO 2017 and 6 shots for COCO 2014. We updated the paper to detail this analysis in Section 4.3 at Page 9, and Section A.1.2 at Page 14.

---

> ### Author Response · Authors · 2023-11-19
> **Response to reviewer 1 (W5-W6)**
>
> **(W5, one-shot vs few-shot) Do prior one-shot object detection methods also follow the few-shot paradigm of fine-tuning the model? If not, what exactly is the difference between the proposed method and one-shot methods?**
>
> One-shot object detection does not follow the finetuning paradigm of few-shot. However, one-shot detection restricts the setting to single-class detection [1], and focuses on designing interaction mechanisms of dense spatial features between the support image and target images. One-shot detection requires separate inference for each class and does not leverage additional support images even if available, as mentioned in the related work [2].
>
> Compared to one-shot detection work, our method focuses on the interaction between class-level prototypes with dense features of target images, instead of interaction of dense spatial features between support and target images. Our method does not require separate inference for each class and can leverage multiple support images.
>
> [1]: One-Shot Instance Segmentation, NeurIPS 2018
>
> [2]: One-Shot Object Detection with Co-Attention and Co-Excitation, NeurIPS 2019
>
>
> **(W6, use dot-product in Eqs)**
>
> We thank the reviewer for this suggestion. We updated the corresponding equations with dot-product.

---

### Author Response · Authors · 2023-11-21
**Code and pretrained models are released on Github anonymously**

We released all code, and pretrained models on COCO and LVIS in https://github.com/anonymous368/devit anonymously.

Please let us know if you have any additional questions or concerns.

---

### Meta-Review · Area_Chair_76iN · 2023-12-06

**Metareview:**

The paper presents a few-shot vocabulary object detector (DE-ViT) built on DINOv2. The detector learns new concepts by seeing a few example images (1, 10 or 30) of each class.
Overall, two reviewers recommend borderline acceptance, two borderline rejection.
+ The paper is well written and easy to understand
+ The papers shows promising results on few-shot detection
- The main claim in the paper "first few-shot detector build on DINOv2" undermines novelty claims. While the statement is true, being the first to build a detector on a specific backbone does not appear very novel by itself. At the same time, it highlights that the broader statement of being the first detection build on image features of a few classes is not true (OVDetr and other methods already have that capability). As such the novelty in the presented paper is limited.
- As pointed out by Reviewer Vm41 (and acknowledged by the authors in their response) the comparison to open-vocabulary detectors is misleading and not apples-to-apples. While the authors revised the paper, the most recent draft still mixes and compares the two settings without any reflection on the fairness of the comparison. The AC considers this a major issue.

Overall, the paper is on a good track, but has flaws that have not been resolved adequately. The AC thus does not see a path to acceptance for this submission.

**Justification For Why Not Higher Score:**

I read the papers, comments, and rebuttal. In the final revision, the reviewers concerns have not been addressed adequately (see above).

**Justification For Why Not Lower Score:**

N/A. Desk reject deadline already passed :)

---

### Decision · Program_Chairs · 2024-01-16

Reject